# L-threonine promotes healthspan by expediting ferritin-dependent ferroptosis inhibition in *C. elegans*

Juewon Kim [1] ✉, Yunju Jo[2], Donghyun Cho[1] & Dongryeol Ryu [2] ✉

The pathways that impact longevity in the wake of dietary restriction (DR) remain still ill-defined. Most studies have focused on nutrient limitation and perturbations of energy metabolism. We showed that the L-threonine was elevated in *Caenorhabditis elegans* under DR, and that L-threonine supplementation increased its healthspan. Using metabolic and transcriptomic profiling in worms that were fed with RNAi to induce loss of key candidate mediators. L-threonine supplementation and loss-of-threonine dehydrogenaseincreased the healthspan by attenuating ferroptosis in a ferritin-dependent manner. Transcriptomic analysis showed that *FTN-1* encoding ferritin was elevated, implying *FTN-1* is an essential mediator of longevity promotion. Organismal ferritin levels were positively correlated with chronological aging and L-threonine supplementation protected against age-associated ferroptosis through the DAF-16 and HSF-1 pathways. Our investigation uncovered the role of a distinct and universal metabolite, L-threonine, in DR-mediated improvement in organismal healthspan, suggesting it could be an effective intervention for preventing senescence progression and age-induced ferroptosis.

Reduced food intake without malnutrition can attenuate age-associated disorders in invertebrate model organisms and mammals, including humans[1,2]. Dietary restriction (DR), defined as the coordinated reduction of intake of all dietary ingredients except vitamins and minerals, extends the lifespan of aging model organisms and improves most health phenotypes during aging[3]. Because of diminished calorie intake during DR, most metabolic processes and metabolic intermediates are reduced, and this decreased metabolic network can induce efficient metabolic activity in many organisms[4,5]. Thus, it is reasonable to postulate that high metabolic efficiency improves many of the physiological and molecular changes involved in age-related autonomic dysfunctions. Therefore, we speculated that increased metabolites during DR interventions might be involved in the attenuation of age-associated decline. To test this hypothesis, we screened for elevated metabolites in DR and a DR-related mutant in *Caenorhabditis elegans*. We identified several metabolic intermediates

whose concentration is altered, of which the most increased was the essential amino acid L-threonine.

Nutrient amino acid imbalance has been investigated with regard of its impact on physiological processes, health, and the regulation of aging. Previous studies reported that imbalanced amino acid levels may contribute to the incidence of adverse effects on aging in a rat model and suggested that reduced intake of a particular protein and amino acids may have essential roles in explaining the benefits of DR. However, the medical literature and mass media often suggest that high protein supplements, particularly those rich in essential amino acids, and having branched-chain amino acids, can overcome age-related sarcopenia, obesity, frailty, and mortality. Recent studies have been in line with this opinion, showing that specific amino acids could positively affect the organismal lifespan and health factors. For example, glycine extended the lifespan of *C. elegans* in a methionine cycle-dependent manner, and glycine supplementation extended the

[1]Basic Research & Innovation Division, Amorepacific R&D Center, Yongin, Korea. [2]Department of Molecular Cell Biology, Sungkyunkwan University School of Medicine, Suwon, Korea. ✉e-mail: kimjw@amorepacific.com; freefall@skku.edu

lifespan of male and female mice[6,7]. Additionally, branched-chain amino acids induced a central neuroendocrine response, thereby extending healthspan[8], facilitating survival, and supporting cardiac and skeletal muscle mitochondrial biogenesis in middle-aged mice[9]. These studies supported the theory that regulation of dietary amino acid levels can enhance healthspan in organisms and provided evidence in support of the further investigation of the effects of characteristic amino acids on senescence and late-life diseases[10,11].

In the case of L-threonine, it was reported that threonine metabolism was a crucial factor for stem cell maintenance and renewal by influencing histone methylation[12–15]. Moreover, as a major component of mucins, threonine plays a conclusive role in the maintenance of mucosal integrity, barrier constitution, and eventually immune function regulation[16–19]. Recently, the effects of threonine on age-associated metabolic pathways, lipid metabolism[20], and methylglyoxal (MGO)-mediated proteohormesis[21] were investigated. In this study, it was revealed that impaired threonine catabolism promoted *C. elegans* healthspan by activating a highly reactive compound in the MGO-mediated proteasome pathway. Despite these initial indications on the importance of threonine, its roles and metabolism in aging remain largely unknown.

If the effect of threonine on aging is conserved across metazoa and beyond, lifespan regulation by altered threonine metabolism should also be conserved in the commonly used aging model, *C. elegans*. We here uncovered the role of threonine and its catabolic enzyme threonine dehydrogenase (TDH) inhibition in *C. elegans* lifespan and healthspan regulation and suggest that it is mediated by ferritin-mediated ferroptosis. The use of defined diets with specific changes in amino acids composition in *C. elegans* could hence provide an experimental system with which to examine the consequences of altered amino acid metabolism and its effects on aging and other physiological functions.

## Results

### Identification of longevity-related metabolites and their effects on *C. elegans* healthspan

To gain perspective into the longevity and change in endogenous metabolites, we sorted dietary restriction (DR)-associated metabolites for their effects on lifespan using the *C. elegans* model. Using a CE-TOFMS and LC-TOFMS metabolic profiling, we detected several metabolic pathways that were significantly altered in worm subjected to the DR and DR-mimic mutant, *eat-2* model (Fig. 1a, Supplementary Data 1). According to previous studies[1,3], DR lowered almost all metabolic processes, because of limited calorie and nutrient intake. In contrast, we discovered that several metabolites were increased under DR and in the *eat-2* mutants compared to the wild type, N2 strain (Fig. 1b). Among the metabolites that were increased, including ribose 5-phosphate, creatinine, and glutathione, several of these metabolites were already recognised to affect longevity, including NAD+[22,23] and metabolites involved in lipid metabolism[24]. Most interestingly, we found that the most elevated amino acid, threonine, delayed the aging process and extended the lifespan of nematodes by 18% (Fig. 1c, Supplementary Fig. 1a, b). The dose-response changes in mean lifespan upon threonine exposure showed that low concentrations enhanced longevity, with 200 μM threonine exhibiting the maximal lifespan extension (Fig. 1d); 200 μM was used for subsequent assays. Threonine not only elongated lifespan, it also improved health-related parameters, average speed, and coordinated body movement (Fig. 1e), and reduced age-related triglyceride (TG) content (Fig. 1f)[25]. Moreover, unlike many other lifespan-extending compounds, there were no significant alterations in body size, pharyngeal pumping rate reflecting calorie intake or brood size caused by bacterial proliferation in the presence of threonine (Supplementary Fig. 1c–f). Importantly, threonine enhanced the activities of major reactive oxygen species (ROS) defence enzymes superoxide dismutase (SOD) and catalase and

increased the against acute oxidative stress with no induction of thermotolerance (Fig. 1g–i, Supplementary Fig. 1g). Additionally, we conducted additional studies with stronger dosages of 10, 20, and 40 mM of threonine to examine the effect on longevity of threonine supplementation with a strictly controlled dose range (Supplementary Fig. 1h–j). We observed that *E. coli* OP50's growth was suppressed by a high threonine dose, which may shorten life duration. With the chemical toxicity of high doses of threonine, the longevity effects of threonine intervention are tightly regulated and limited by calorie restriction-like biological processes (Supplementary Fig. 1k–m). We therefore believed that threonine concentrations beyond 10 mM are unlikely to be encountered in nature outside of experimental settings like genetically modified *E. coli*[26], whereas the suitable dose range of threonine in this investigation (100–400 μM).

Collectively, the DR-related metabolite threonine increased the healthspan by the induction of antioxidative protection. It was furthermore interesting to note that previous studies on anti-aging compounds and metabolites, which reported DR-mimic activity, including syringaresinol[27], metformin[28], and α-ketoglutarate[29], also increased the threonine content in *C. elegans* (Fig. 1j). This indicates that threonine may act as a key amino acid in the DR-mediated longevity and as a primary metabolite for healthspan extension.

### Threonine was downregulated during aging and reduced catabolism related to DR

Based on the threonine metabolic pathway (KEGG_cel00260 and WormFlux_c00188), we attempted to search for age-related changes of metabolic enzymes and metabolites involving threonine (Fig. 2a). Threonine is only utilized by three metabolic reactions in worms (WormFlux c00188). Threonine aldolase, also known as R102.4, governs both the generation of threonine from glycine and acetaldehyde and the conversion of threonine to those two compounds. Threonine is deaminated by the enzyme K01C8.1, which produces 2-oxobutanoate and NH3 from threonine. The other enzyme is threonine dehydrogenase (TDH), also known as F08F3.4; it converts threonine into 2-amino-3-oxobutanoate, H, and NADH. We observed an age-dependent decline of threonine content (Fig. 2b). Furthermore, DR and also *eat-2* animals exhibited decreased glycine and homoserine, which are precursors of threonine synthesis, and increased threonine without conversion (Figs. 1b, 2c). According to the Venn analysis based on public datasets, threonine metabolising enzymes, R102.4 (Threonine aldolase), threonine dehydrogenase (TDH)/F08F3.4, and threonine deaminase/K01C8.1 were located at the intersection of differentially expressed genes (DEG) in the DR treatment (GSE119485, padj <0.05 in edgeR, Deseq2)[30] and the *eat-2* worm DEG (GSE146412, padj <0.05 in edgeR, Deseq2)[31] (Fig. 2d, Supplementary Data 2). In the DR and *eat-2* mutant, threonine-metabolising enzyme R102.4 expression was upregulated (Fig. 2e), and the expression of threonine catabolic enzyme F08F3.4 was downregulated (Fig. 2f). Compared to long-lived worms, *R102.4* expression was diminished in day 5 nematodes (Fig. 2h), and *F08F3.4* expression increased with age (Fig. 2i). Under both conditions, the expression level of K01C8.1 did not change (Fig. 2g, j). Increased synthesis (R102.4), together with reduced catabolism (TDH/F08F3.4) of threonine, caused an increase in threonine abundance, which potentially acted as a longevity metabolite in *C. elegans*.

### Threonine catabolism enzyme impairs the promotion of the healthspan of *C. elegans*

Based on increased threonine levels, we applied RNA interference (RNAi) against *tdh*/F08F3.4 (TDH RNAi) to N2 nematodes. We attained a mean lifespan extension of 36.1% under the suppression of *tdh*/F08F3.4 expression (Fig. 2k, Supplementary Fig. 2a–c). With the concomitant increased amount of threonine (Fig. 2l), TDH RNAi stimulated locomotion (Fig. 2m) without alteration of physical

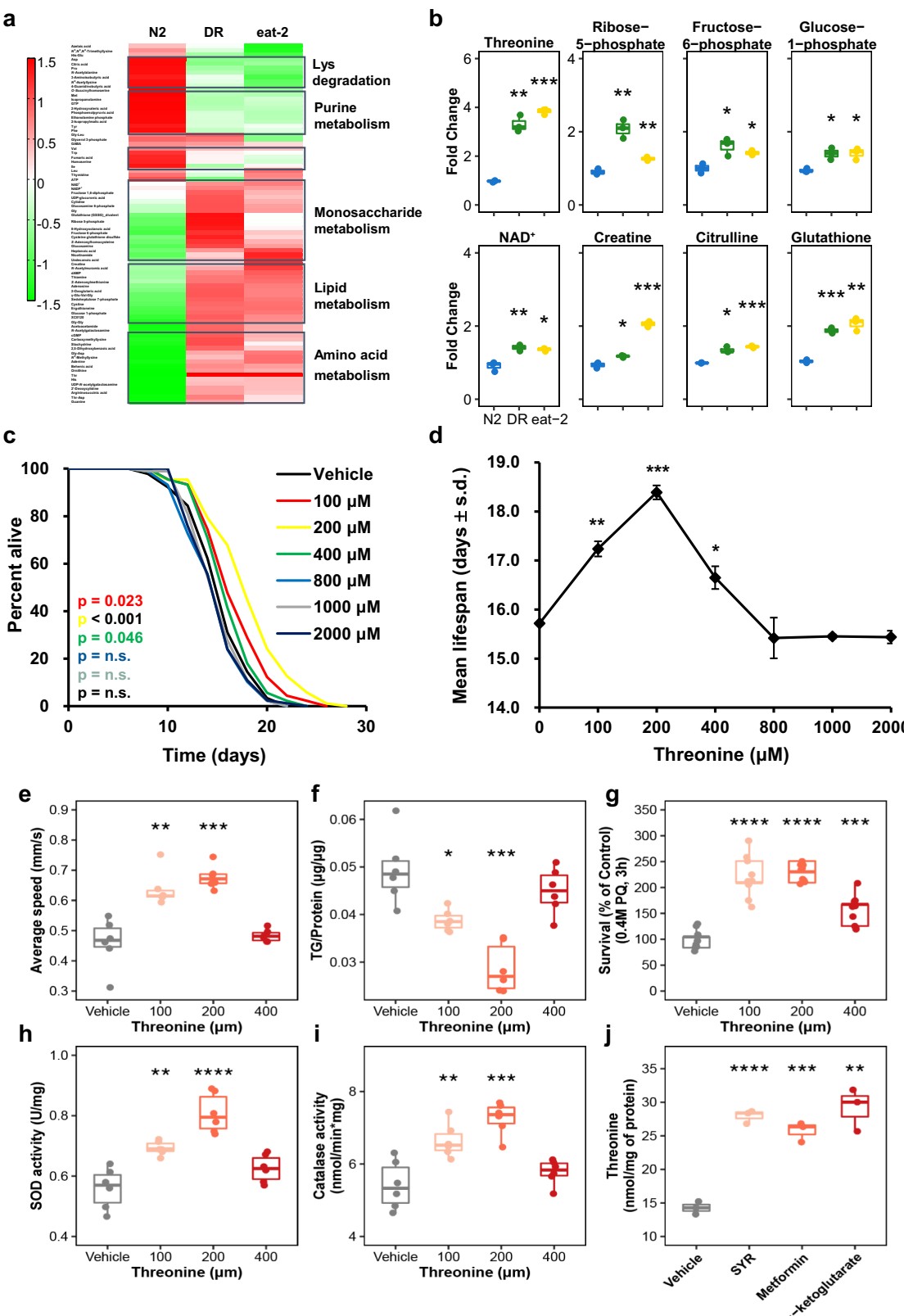

body size, pumping rate, or progeny (Fig. 2n, Supplementary Fig. 2d, e). Additionally, similar to the threonine treatment condition, TDH RNAi lowered TG contents (Fig. 2o) and improved anti-oxidative defence (Fig. 2p–r) with no difference in resistance against thermal stress or bacterial growth (Supplementary Fig. 2f, g). We next determined whether blocking threonine generation affected aging parameters. The expression of the threonine-

metabolising R102.4 enzyme was increased in the dietary restricted *eat-2* mutants and decreased in early-adult day 5 worms (Fig. 2e, h). We used R102.4 RNAi to inhibit threonine production. In contrast to the observations for increased threonine levels (Figs. 1c–i, 2k–r), R102.4 RNAi shortened lifespan by 16.8% and reduced threonine (Supplementary Fig. 3a–e), further supporting threonine's regulatory action in the modulation of healthspan

Fig. 1 | **Threonine is increased in long-lived nematodes and its supplementation improves fitness and healthspan. a** Differentially altered metabolites, as quantified by metabolome analysis under wild-type N2 conditions, dietary restriction (DR), or the *eat-2* worms. Increased and decreased metabolites were examined by hierarchical clustering to determine the pathway and metabolites changed under the long-lived condition. **b** Significantly elevated metabolites in DR and *eat-2* compared to N2 ($**p < 0.01$ versus N2, two-tailed *t*-test, false discovery rate (FDR) < 0.05, $n = 3$ worm pellets). **c** Effects of L-threonine (100–2000 µM) versus the vehicle (black) on lifespan (*p*-value listed, respectively, log-rank test); color coding assigned to all subsequent panels. **d** Mean lifespan of threonine treatment by concentration is depicted (mean ± s.d, $*p < 0.05$, $**p < 0.01$, and $***p < 0.001$, log-rank test, $n = 83–96$ worms with 3 independent assays). **e–i** Effects of threonine versus the vehicle in animals regarding (**e**) average speed of locomotion ($*p = 0.048$ and $**p = 0.005$ versus the vehicle group, $n = 10–15$ worms with 3 independent assays), (**f**) triglyceride (TG) content ($*p = 0.048$ and $**p = 0.006$ versus the vehicle group, $n = 3$ worm pellets), (**g**) oxidative stress resistance ($***p < 0.001$ versus the vehicle group, $n = 20$ worms with 9 independent measurements each), (**h**) superoxide dismutase (SOD) activity ($*p = 0.038$ and $**p = 0.012$ versus the vehicle group, $n = 3$ worm pellets), and **i** catalase activity ($*p = 0.042$ and $**p = 0.008$ versus the vehicle group, $n = 3$ worm pellets). **j** Previously reported DR-mimetic reagents also increased threonine in Day 10 worms ($***p < 0.001$, $n = 3$ worm pellets each). **e–g** P values were determined by two-sided *t*-test. FDR < 0.05. Error bars represent the mean ± s.d. Source data are provided as a Source Data file.

(Supplementary Fig. 3f–o). In summary, impairment of threonine metabolism by RNAi clones targeting two different enzymes of threonine metabolism altered the lifespan of wild-type worms. Large amounts of threonine occurred in long-lived *C. elegans* compared to the wild-type N2 nematodes and upregulated threonine contents induced healthspan prolongation, suggesting that increasing threonine enhances longevity of this model organism.

### Threonine-mediated longevity depends on oxidative stress-response transcription factors DAF-16 and HSF-1

To determine the molecular mechanisms of the longevity effects of threonine, we performed RNA sequencing (RNA-seq) on nematodes exposed to *tdh*/F08F3.4 RNAi for 5 d. We found 2171 and 2110 DEGs, respectively, which were upregulated and downregulated in comparison with control RNAi (Fig. 3a). With these remarkable changes in the transcriptome (Fig. 3b, c), impaired *tdh*/F08F3.4 expression resulted in a metabolic shift and alterations in senescence-regulatory pathways (Supplementary Fig. 4a). All DEGs were combined, and using clustered expression profiles for DEGs of DR and *eat-2* worms compared to those of the wild type nematodes, we identified 23 upregulated and 13 downregulated genes (Supplementary Fig. 4b, c). Focusing on the transcription factors of these DEGs and and how they may regulate lifespan, we next tested whether *tdh*/F08F3.4 has an epistatic interaction with DR, insulin/IGF-1, or stress-response regulators. Similar to the results for increased threonine and decreased *tdh*/F08F3.4 levels in DR and *eat-2* mutants (Figs. 1b, 2f), TDH RNAi and threonine application failed to further extend the lifespan of *eat-2(ad465)*, which was different from N2 nematodes (Fig. 3d, Supplementary Figs. 4d, e, 5a–c). Interestingly, threonine catabolism impairment, or threonine itself, interacted with *daf-16*, although its upstream transcription factor *daf-2* did not (Fig. 3e, f, Supplementary Fig. 4f–i). This requirement of *daf-16* was specific because the lifespan of the even longer-lived insulin/IGF-1 receptor *daf-2* mutant worms was further prolonged by TDH RNAi and threonine (Supplementary Fig. 5d–i). It may be because *daf-2* mutant worms have been described to showed decreased threonine levels[32]. Lastly, and according to a previous study with glycine-C-acetyltransferase (GCAT) that metabolizes the 2-amino-3-ketobutyrate product of the TDH reaction[21], transcriptional stress-response regulator HSF-1 was also considered. The higher antioxidative enzyme activity and oxidative stress resistance in threonine-treated animals suggested that threonine-mediated longevity may involve a similar context to that initiated by stress stimulation. Furthermore, we confirmed that lifespan extension effects of both TDH RNAi and threonine intervention were fully negated in *hsf-1(sy441)* worms, which lack the transactivation domain of the protein (Fig. 3g, Supplementary Figs. 4j, k, 5j–l). These findings demonstrated epistatic interplay of threonine with *daf-16* and *hsf-1*. In addition we confirmed that the nuclear translocation and increased expression of *daf-16::gfp* or *hsf-1::gfp* was intensely stimulated by *tdh*/F08F3.4 RNAi and additional threonine (Fig. 3h–k).

### FTN-1 expression is required for threonine-mediated longevity

Based on these results, we selected six and two commonly up- and downregulated genes, respectively, which are transcribed by the transcription factors DAF-16 or HSF-1, from the intersecting genes of *tdh*/F08F3.4 RNAi, DR, and *eat-2* worms (Supplementary Data 3). All of these genes were individually targeted by their respective RNAi clone with the exception of genes already examined for threonine metabolism, R102.4 and *tdh*/F08F3.4. We investigated lifespan changes induced by RNAi of these four genes: glutathione *S*-transferase, *gst-19* (Fig. 4a, Supplementary Fig. 6a, b); inferred organismal homeostasis of metabolism variant, T12D8.5 (Fig. 4b, Supplementary Fig. 6c, d); *Caenorhabditis* bacteriocin, *cnc-2* (Fig. 4c, Supplementary Fig. 6e, f), and ferritin, *ftn-1* with *tdh*/F08F3.4. The RNAi revealed that only *ftn-1*/C54F6.14 exhibited an interaction with the impaired *tdh*/F08F3.4 expression (Fig. 4d, Supplementary Fig. 6g, h). *tdh*/F08F3.4 RNAi and threonine were furthermore unable to additively extend the lifespan under the *ftn-1* null mutant, *ftn-1(ok3625)* (Fig. 4e, f, Supplementary Fig. 6i–l), indicating that the conserved iron-cage protein, ferritin 1 (FTN-1) is essential for threonine-mediated longevity.

To determine the significance of FTN-1 to threonine-related longevity, we measured *ftn-1* expression with physiological aging and in N2 worms fed with *tdh*/F08F3.4 RNAi. As previously reported[33], *ftn-1* expression increased with age, but more importantly, its expression was further elevated by the reduced expression of *tdh*/F08F3.4, especially at early adult stages (Fig. 4g, h). Thus, FTN-1 is required for the longevity advantage of threonine. Given the interaction of *ftn-1* expression with several genes, we exposed DR and *eat-2* worms to the target gene RNAi, with control RNAi serving as control. Although von Hipple-Lindau protein 1 (*vhl-1*), hypoxia-inducible transcription factor (*hif-1*), and prolyl hydroxylase *egl-9* had no effect on *ftn-1* expression, *daf-16* and *hsf-1* did affect elevated *ftn-1* expression in DR and *eat-2* worms, compared with the wild-type nematodes (Supplementary Fig. 7a, b). This may have occurred because of the specific regulation of *ftn-1* in the threonine-mediated longevity condition. Because *ftn-1* expression is modulated by oxygen concentration, *ftn-1* was regulated differently under normoxia, hypoxia, and hyperoxia conditions using specific mutations of the target genes (Supplementary Fig. 7c–e). Similar to the DR and *eat-2* animals, *daf-16* and *hsf-1* were necessary to induce *ftn-1* expression by *tdh*/F08F3.4 RNAi, indicating threonine-induced longevity is impacted by increasing organismal levels of FTN-1.

### FTN-1 mediates threonine-dependent ferroptosis alleviation

We next hypothesized that altered iron content might contribute to the long-lived phenotype. Similar to humans, *C. elegans* FTN-1 is induced by iron, and accumulated reactive ferrous iron results in a non-apoptotic form of cell death, ferroptosis[34]. Quantifying the ferrous iron ($Fe^{2+}$, redox-active) / ferric iron ($Fe^{3+}$, stable) concentrations following *tdh*/F08F3.4 RNAi or threonine addition also suggested a possible global diminution in organismal age-dependent ferroptosis (Fig. 4i, Supplementary Fig. 7f). Decreased levels of ferrous iron may result from expanded iron retention by FTN-1 and result in inactivating

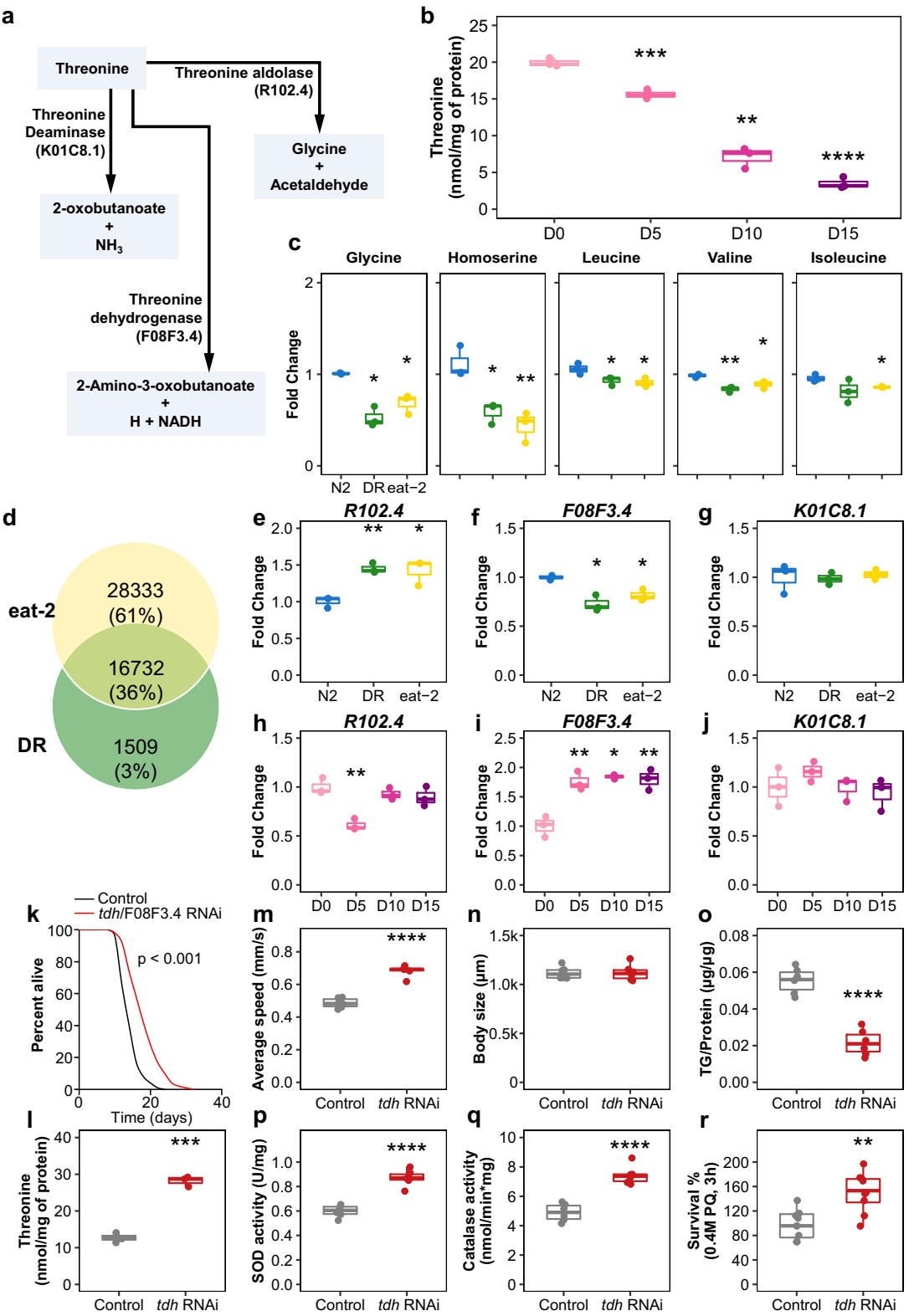

ferroptosis. Moreover, this redox-active iron ratio ($Fe^{2+}/Fe^{3+}$) was also dependent on *daf-16* and *hsf-1*, similar to the regulation of *ftn-1* expression in DR or *eat-2* nematodes (Supplementary Fig. 7g, h). In the ferroptosis pathway, ferrous iron accumulates over the adult lifetime of *C. elegans*[35] and exists as abundant reactive ferrous iron, which could generate massive damage through ROS, eventually inducing ferroptosis and shortening the lifespan[36,37]. Hence, with the changes accruing

during physiological aging, we detected in worms fed with *tdh*/F08F3.4 RNAi or with supplemental threonine an associated decrease in ROS levels in whole nematode lysates, particularly during the late adult stage (Fig. 4j, Supplementary Fig. 7i). Additionally, we confirmed that the absence of *daf-16* or *hsf-1* failed to reduce ROS levels in DR and *eat-2* worms, analogous to *ftn-1* expression (Supplementary Fig. 8a–c). As a substrate for the main regulator of ferroptosis glutathione

**Fig. 2 | Threonine declines with age and threonine catabolism disorder enhances lifespan and healthspan. a** Threonine is metabolized by threonine dehydrogenase (TDH) to 2-amino-3-ketobutylate, which then either spontaneously decarboxylates into aminoacetone or is metabolized by the GCAT enzyme using CoA as a co-substrate to form glycine and acetyl-CoA. **b** Threonine abundance decreased with age (***$p < 0.001$ versus the first time point, $n = 3$ worm pellets). **c** Glycine and homoserine decreased in DR or *eat-2* nematodes (***$p < 0.001$ versus N2 group, $n = 3$ of 5-day-old-worm pellets). **d** Venn analysis of DR and *eat-2* mutant using publicly available datasets. Intersection of differentially expressed genes (DEGs) of DR ($p < 0.05$ in edgeR, Deseq) with *eat-2* mutant DEGs compared to the wild-type N2 ($p < 0.05$). **e–g** Among the common DEGs, the expression of enzymes that were related with threonine metabolism were determined as (**e**) *R102.4* (**$p < 0.01$ and ***$p < 0.001$ versus the N2 group, $n = 3$ of 5-day-old-worm pellets), (**f**) *TDH/F08F3.4* (***$p < 0.001$ versus the N2 group, $n = 3$ of 5-day-old-worm pellets),

and (**g**) *K01C8.1* ($p > 0.05$, $n = 3$ of 5-day-old-worm pellets). The expression of these enzymes was measured with age as (**h**) *R102.4* (**$p < 0.01$ versus the first time point, $n = 3$ worm pellets), (**i**) *TDH/F08F3.4* (**$p < 0.01$ and ***$p < 0.001$ versus the first time point, $n = 3$ worm pellets), and (**j**) *K01C8.1* ($p > 0.05$). **k** Lifespan analysis of *tdh/F08F3.4* RNAi (mean ± s.d, $p < 0.001$, log-rank test, $n = 88$–$96$ worms with 3 independent assays) versus control RNAi in N2 worms. **l–r** Effects of *tdh/F08F3.4* RNAi versus control RNAi for (**l**) threonine contents (***$p < 0.001$, $n = 3$ worm pellets), (**m**) average speed (***$p < 0.001$, $n = 10$–$15$ worms with 3 independent assays), (**n**) body length ($p = 0.785$, $n = 6$), (**o**) triglyceride (TG) content (***$p < 0.001$, $n = 3$ worm pellets), (**p**) superoxide dismutase (SOD) activity (**$p = 0.007$, $n = 3$ worm pellets), (**q**) catalase activity (**$p = 0.002$, $n = 3$ worm pellets), and (**r**) oxidative stress resistance (***$p < 0.001$, $n = 20$ worms × 9 measurements each). **b–c**, **e–j**, and **m–r** $P$ values were determined by two-sided *t*-test. FDR < 0.05. Error bars represent the mean ± s.d. Source data are provided as a Source Data file.

peroxidase 4 (GPX4), glutathione (GSH) deficiency is coupled to ferrous iron accumulation in nematodes, and both occur to stress cells leading to ferroptosis[35]. To systematically examine this, we determined whole GSH levels in worm lysates after *tdh*/F08F3.4 RNAi or threonine feeding at time points during aging. There was an age-related decline in GSH levels, and addition of increasing amounts of threonine prevented GSH depletion (Fig. 4k, Supplementary Fig. 7j) and increased the GSH level in DR and *eat-2* worms, which was modulated by *daf-16* or *hsf-1* as assayed by the target gene RNAi (Supplementary Fig. 8d–f). Redox-active iron-dependent lipoxygenases and iron-catalysed peroxyl radical-mediated autoxidation has been shown to induce lipid peroxidation, which mediates ferroptosis[38]. We examined the toxic lipid peroxidation end-product malondialdehyde (MDA) to assay the ferroptosis level[39], *tdh*/F08F3.4 RNAi or threonine feeding reduced MDA levels in an age-dependent manner (Fig. 4l, Supplementary Fig. 7k). Similarly, the MDA level was lowered dependent on *daf-16* or *hsf-1* in DR and *eat-2* nematodes (Supplementary Fig. 8g–i). Additionally, we assessed the specificity of ftn-1 to ferroptosis and confirmed that ferrostatin-1 treatment extends lifespan in an ftn-1-dependent manner (Supplementary Fig. 7l–q). However, could not recover hydrogen peroxide and MDA, as well as GSH levels, with ftn-1 RNAi (Supplementary Fig. 7r–t). The ferrostatin-1 effects of ferroptosis inhibition were absolutely associated with ferritin activity. This, together with the results summarized above, suggested that increasing threonine interventions attenuate compromised ferritin capacity, inhibiting redox-active iron-mediated ferroptosis. To test this hypothesis, we determined the inhibitory effects of threonine elevation treatment against ferroptosis stimulated by introducing *ftn-1*/C54F6.14 RNAi. As expected, with recovery of wild-type *ftn-1* expression (Fig. 4h), increasing threonine application reduced ferroptosis markers in *ftn-1* mutant nematodes. Treatment of *tdh*/F08F3.4 RNAi or threonine supplementation decreased the redox-active iron ratio, ROS levels, GSH levels, and also MDA levels in a whole nematode lysates in FTN-1-dependent manner (Fig. 4i–l). Furthermore, we confirmed the suppressive effects of increasing threonine treatment against ferroptosis stimulated by using acute depletion of GSH by diethyl maleate (DEM)[40] or a glutamate/cystine antiporter suppressor, erastin[41]. The results provide convincing evidence for the anti-ferroptosis activity of *tdh*/F08F3.4 RNAi or threonine supplementation against exposure to ferroptosis-inducing agents as reflected by reduced ROS, increased GSH, and reduced MDA levels (Fig. 4m–o). When restoring the cellular GSH level by threonine interventions, we measured the highly reactive by-product of glycolysis, MGO, with its scavenger, GSH. MGO is one of the most representative cell-permeant precursors of advanced glycation end-products (AGEs), which are associated with several age-related diseases. The MGO/GSH ratio was lowered with manipulations that increased threonine levels; indicating, it was beneficial for longevity (Supplementary Fig. 8l–n). Furthermore, the MGO/GSH ratio increased with aging, and long-lived DR and *eat-2* worms showed a greatly decreased MGO/GSH ratio and a concordantly increased

lifespan (Supplementary Fig. 8o, p). In addition, glycine, the metabolite of threonine and also a slightly increasing amino acid with *C. elegans* aging, administration could not affect the ftn-1 expression with aging, $Fe^{2+}/Fe^{3+}$ ratio, and MDA levels, although it slightly reduced hydrogen peroxide and increased glutathione levels under ferroptosis-induced conditions (Supplementary Fig. 8q–s). These results suggest that ferroptosis inhibition via FTN-1 activation is specific for threonine supplementation. On the other hand, glycine affects *C. elegans*' lifespan by different mechanisms with ferroptosis, which could reduce ROS.

Furthermore, to expand our understanding of threonine metabolism in mammals, we analysed the publicly available data set of the BXD mouse genetic reference population (http://www.genenetwork.org/) (PMID: 22939713, 34552269, 33472028). First, we looked for any trait indicating threonine quantity in the BXD mice population's phenotypes and then discovered a record of midbrain threonine concentration. The concentration of threonine in the midbrain varied across BXD mice strains (Fig. 5a). Next, we analysed correlations between the midbrain threonine concentration and all accessible phenotypes of the BXD mouse reference population. Interestingly, the concentration of threonine in the midbrain was positively correlated with lifespan features (Fig. 5b, c). To see the effect of the midbrain concentration on the lifespan of the BXD mouse reference population, we estimated survival rate with Kaplan–Meier estimator using the survival record of the BXD strains under high fat diet. Survival rates were increased in BXD mice strains with a higher midbrain threonine concentration (Fig. 5d), which is commensurate with the results observed in C. elegans. Although it is difficult to apply the current study of threonine metabolism in model organisms that have multiple pathways for threonine catabolism to humans that lack functional threonine aldolase and threonine dehydrogenase enzymes. However, mice have both functional threonine aldolase and threonine dehydrogenase similar with *C. elegans*. With this data, researchers may be able to better understand the role of threonine in age-related changes in higher organisms. In the current study, we focused on threonine as a commonly increased metabolite in DR-mediated long-lived worms. Blocking threonine dehydrogenase or supplementation with threonine led to increased organismal threonine content. Increased threonine content led to the activation of FTN-1 and the antioxidant system, which attenuated ferroptosis, and mediated lifespan and healthspan extension in *C. elegans* through a DAF-16 and HSF-1 dependent mechanism (Fig. 6).

## Discussion

We measured commonly increased metabolites that may affect longevity in DR-mediated long-lived animals through a metabolomics approach. Among the macronutrients, we identified threonine as most increased amino acid, despite previous studies showing that proteins or amino acids restriction is the the most potent singular dietary intervention beneficially impacting on longevity and age-associated diseases[8,42]. Besides its role in protein metabolism, threonine also

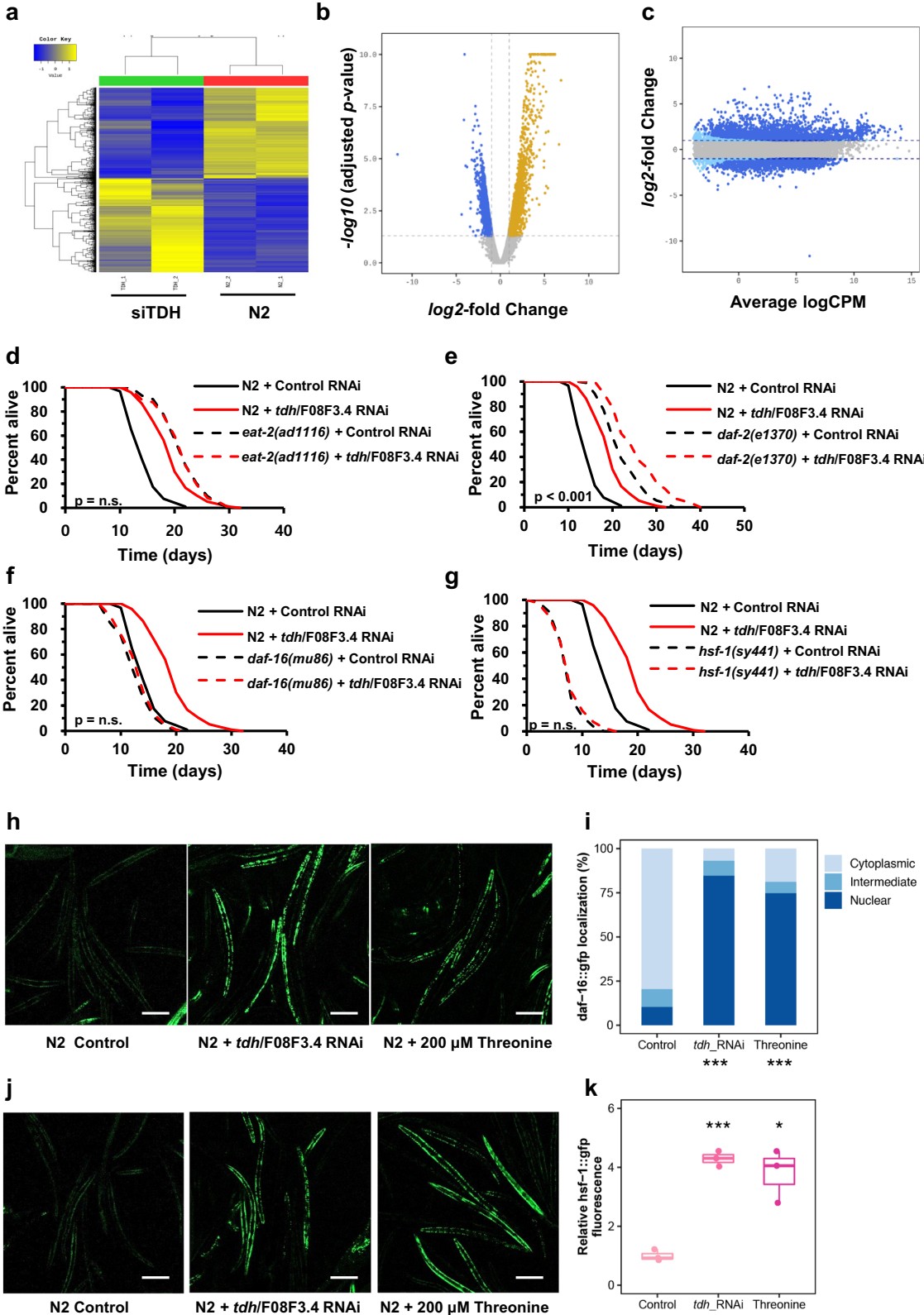

impacts organismal maintenance processes, such as the synthesis of immune proteins and production of intestinal mucus layers[19,43,44]. According to whole-genome sequencing analysis of semi-supercentenarians in Italy, glycine, serine, and threonine metabolism were identified as the top-tier pathways involved in longevity[45]. Furthermore, a recent multi-omics study in mice revealed that the threonine pathway was involved in longevity and related molecular mechanisms[46]. Based on previous reports, almost all amino acids, except glycine and aspartate, decreased with aging in *C. elegans* and the level of threonine metabolizing enzyme, R102.4 restored in day 9 worms[7,47]. We speculated that the slightly increased glycine level can be converted to threonine via R102.4 in adult animals. Together these studies indicated that threonine or its metabolites may affect organismal health- and life-span.

**Fig. 3 | Lengthening of health span by _tdh/F08F3.4_ RNAi and threonine supplement is mediated by DAF-16 and HSF-1-dependent metabolic changes.**
**a** Gene clusters whose expression is regulated by _tdh/F08F3.4_ RNAi versus the wild-type N2 are represented by hierarchical clustering analysis (Euclidean distance, complete linkage). **b** DEGs in _tdh/F08F3.4_ RNAi compared to the N2 visualized with a volcano plot. An adjusted _p_ value < 0.05 and |log$_2$ (_tdh/F08F3.4_ RNAi / N2 ratio)| >2 were applied to define significantly up- or downregulated genes (blue and yellow dots, respectively) or genes without change in expression (grey dots). **c** DEGs as quantified by transcript levels by deep sequencing analysis under treatment with _tdh/F08F3.4_ RNAi; grey dots represent no differential regulation; dark blue and light blue dots depict DEGs according to edgeR analysis (dark blue: |FC| ≥ 2 with

_p_ < 0.05; light blue |FC| ≥ 2). **d–g** Lifespan assay of _tdh/F08F3.4_ RNAi (red-dashed) versus control RNAi (black-dashed) for **d** _eat-2(ad465)_ (_p_ = 0.786), (**e**) _daf-2(e1370)_ (_p_ < 0.001), (**f**) _daf-16(m26)_ (_p_ = 0.627), and (**g**) _hsf-1(sy441)_ (_p_ = 0.492) mutant worms with survival curves of N2 (Bold-line). **d–g** _P_ value determined by two-sided log-rank test. **h** Expression and localization (cytoplasm/intermediate/nuclear) of DAF-16::GFP under control conditions, after exposure to _tdh/F08F3.4_ RNAi or 200 μM threonine treatment for 5 d, and **i**, localization quantifications (***_p_ < 0.001, $\chi^2$ test). **j** Expression of HSF-1::GFP at control, _tdh/F08F3.4_ RNAi, or 200 μM threonine, and **k** corresponding quantifications (***_p_ < 0.001, $\chi_2$ test). Scale bars, 200 μm. Error bars represent the mean ± s.d. Source data are provided as a Source Data file.

Impairing threonine catabolism has been reported to promote the healthspan of _C. elegans_. A past study concentrated on investigating the action of glycine C-acetyltransferase (GCAT), which is involved in the conversion of the intermediate 2-amino-3-ketobutyrate to glycine and its impairment resulting in an increment in MGO-mediated proteohormesis[21]. Here, we focused on threonine content itself. We investigated the role of threonine and enzymes involved in its metabolism, which directly influence threonine abundance. First, we determined whether threonine could affect _C. elegans_ lifespan or if its increased abundance was only a consequence of longevity. Low-dose threonine enhanced lifespan and healthspan-related parameters with improved antioxidative enzyme activities. Because DR-mediated longevity has been correlated with reduced protein and amino acid intake, these effects raise an intriguing question about how certain amino acids have the potential to promote healthspan. Although many studies have reported the benefits of amino acid restriction, recent research has provided clues about the longevity advantages of specific amino acids in many organisms. Supplementation of the single amino acid glycine extended lifespan of _C. elegans_[7], _Drosophila_[48], and mice[6] by attenuation of the progression of dystrophic pathologies. In addition, branched-chain amino acids, a group of essential amino acids, such as leucine, isoleucine, and valine, could impact health and lifespan by affecting amino acid balance[8,9,49]. Incited by these findings, we investigated the merits of threonine on fitness and lifespan. Based on published RNA-seq data, the threonine metabolizing enzyme, R102.4, was shown to be upregulated in DR-related mutant _eat-2(ad465)_ worms and TDH/F08F3.4 was identified amongst the DEGs in the long-lived mutants, _daf-2(e1370)_ and _eat-2(ad465)_[7,32]. Both threonine synthesis using glycine and homoserine and inhibition of the threonine catabolic enzyme TDH increase threonine contents in DR-mediated long-lived worms.

We showed that threonine upregulates ferritin, FTN-1, expression. Although the total amount of ferritin protein is increased, the iron storage capacity of ferritin is compromised with age and ablated with senescence[35,50]. Escape of iron from ferritin raised cellular ferrous load in the aging worms and this redox-active iron increased the generation of ROS via of its indiscriminate reaction with endogenous cellular peroxides by the Fenton reaction[51]. Together with significant elevation in iron levels, a changing redox environment favouring unrestrained oxidative activity of Fe$^{2+}$ is a remarkable candidate that might explain age-associated ferroptosis and diseases. Indeed, ceruloplasmin-deficient mice that lack the capacity to oxidize Fe$^{2+}$ to Fe$^{3+}$ exhibit extreme age-dependent neurodegeneration[52]. We assumed that threonine increases healthspan by attenuating age-related ferroptosis. Increasing threonine levels upregulate FTN-1 renewal through a mechanism involving the transcription factors DAF-16 and HSF-1, subsequently reducing the hallmarks of ferroptosis, the redox-active iron level, and blocking the generation of ROS. Moreover, we detected ferroptosis-related factors, highly reactive aminoacetone, a precursor of MGO, found to induce dose-dependent ferrous iron release from ferritin[53] decreased in threonine-associated longevity.

According to chromatin immunoprecipitation sequencing (ChIP-seq) results, it has been reported that human ferritin, ferritin heavy

chain 1 (FTH-1), expression is induced by the activation of the human homolog of DAF-16, FoxO3[54]. This indicates that ferritin expression might also be regulated by threonine levels in humans through the activation of the FoxO3 transcription factor. Ferritin imbalance provokes diverse health disorders and diseases, such as iron-deficiency anaemia, hypothyroidism, celiac disease in the case of deficiency, adult-onset Still's disease, porphyria, and an acute inflammatory reaction in some cases of COVID-19[55,56]. These observations and manifestations associated with ferritin homeostasis and its turnover are crucial factors to human health and organismal energy balance[57]. So far most of the positive effects of threonine are linked to enhanced immunity, through mucus formation[18,43], and stimulation of pluripotency of embryonic stem cells[12–15]. Here, we propose another mechanism through which threonine can achieve health- and lifespan benefits in _C. elegans_ through the attenuation of cellular ferroptosis. Our work should stimulate additional studies to evaluate whether these effects on fitness and longevity translate to mammals including humans.

## Methods

### Nematodes strains and maintenance
_C. elegans_ strains were maintained on Nematode Growth Media (NGM) agar plates at 20 °C using OP50 bacteria as a food source. The following strains were used (strain, genotype): Bristol N2, wild type; DA1116, _eat-2(ad1116)_II; CB1370, _daf-2(e1370)_III; CF1038, _daf-16(mu86)_I; PS3551, _hsf-1(sy441)_I; TJ356, zIs356 [daf-16p::daf-16a/b::GFP + rol-6(su1006)]; OG532, _hsf-1(sy441)_I; drSi13II [hsf-1p::hsf-1::GFP::unc-54 3'UTR + Cbr-unc-119(+)]; RB2603, _ftn-1(ok3625)_V. They were all obtained from the Caenorhabditis Genetics Center (CGC). All strains are listed in Supplementary Table 1.

### Worm Synchronization and food preparation
To obtain synchronized nematode, six unmated hermaphrodites were placed on an NGM plate with sufficient food. After 4 days, lysed F1 progenies that were selfing hermaphrodites yielded synchronized hermaphrodite eggs. The obtained eggs were grown on NGM plates. OP50 bacteria were cultured at 37 °C in LB medium for up to 12 h. The OP50 concentration was calculated by a combination of counting the colonies and measuring the OD at a wavelength of 600 nm. The OP50 was centrifuged, resuspended and diluted in H$_2$O at bacteria concentrations of $10 \times 10^{11}$ bacteria ml$^{-1}$ or $10 \times 10^9$ for the dietary restriction (DR) experiments until day 4 of feeding beginning at day 1 of adulthood, same condition with previous study which of RNAseq data used in this study (GSE119485).

### Metabolites extraction and metabonomic profile of _C. elegans_
CE-TOFMS and LC-TOFMS-based metabonomic profile of frozen worm pellets was performed (Human Metabolome Technologies, Yamagata, Japan). The wild type N2, DR treatment, or _eat-2_ mutant one-day-old animals were raised over a period of 4 days. Frozen worm samples were mixed with 800 μl of 50% acetonitrile (v/v) containing internal standards (20 μM for cation measurement and 5 μM for anion measurement), homogenized in the presence of

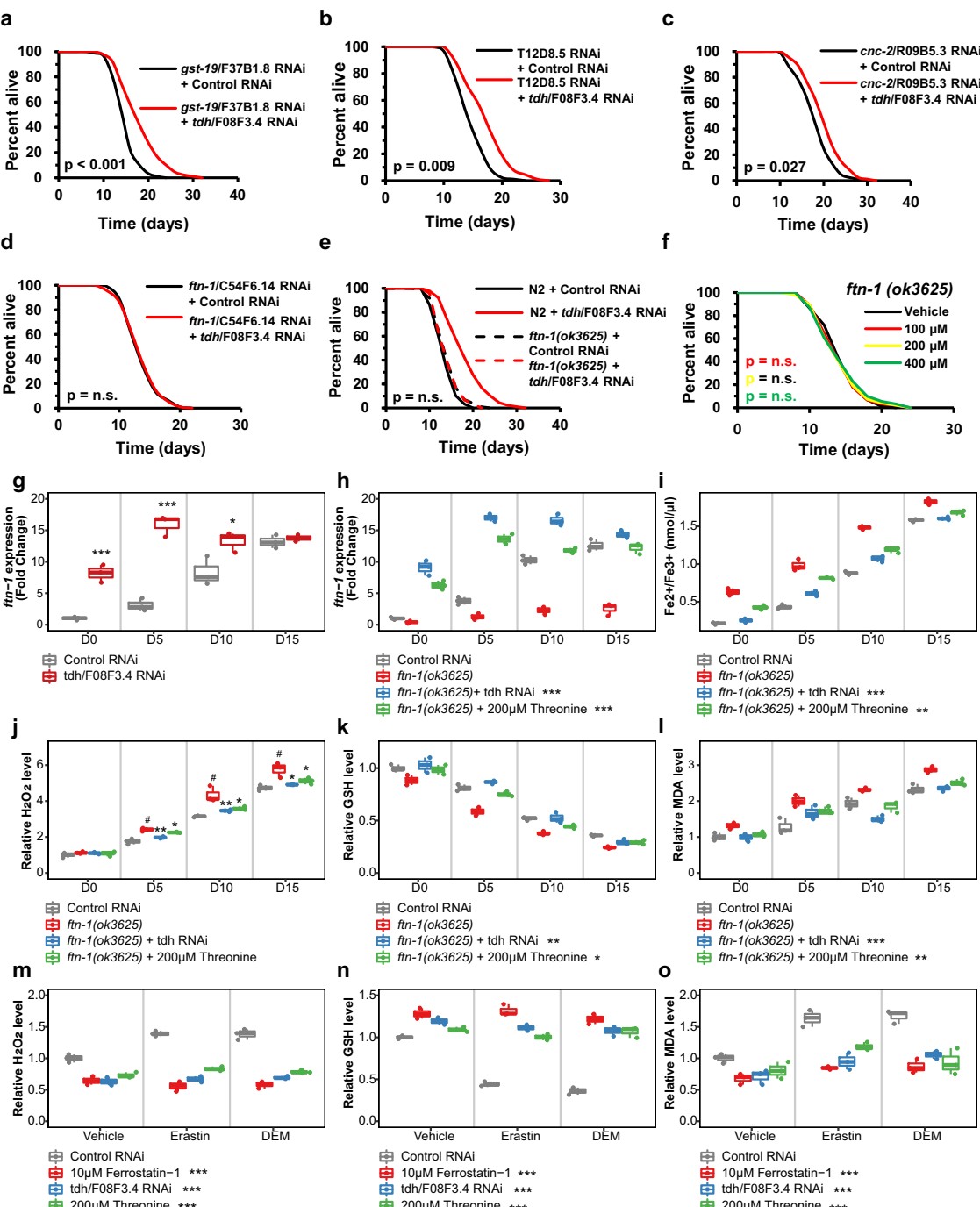

**Fig. 4 | Increasing threonine manipulations upregulate expression of ferritin protein, FTN-1, and FTN-1-mediated ferroptosis suppression. a**–**d** Lifespan assay of *tdh/F08F3.4* RNAi (red) versus control RNAi (black) with transcripts of DAF-16 or HSF-1 among the commonly regulated genes of *tdh/F08F3.4* RNAi, DR, and *eat-2* (by Venn analysis, Supplementary Data 3), (**a**) *gst-19/F37B1.8* RNAi, (**b**) *T12D8.5* RNAi, (**c**) *cnc-2/R09B5.3* RNAi, and **d** *ftn-1/C54F6.14* RNAi worms. **e** Lifespan assay of *ftn-1(ok3625)* with *tdh/F08F3.4* RNAi (red-dashed) versus control RNAi (black-dashed) compared with survival curves of N2 (bold-line). **f** Effects of threonine (100, 200, and 400 μM) versus the vehicle (black) on lifespan of the *ftn-1(ok3625)* mutant. *P* value and lifespan assay data summarized in Supplementary Table 1. **g** Expression of *ftn-1* under *tdh/F08F3.4* RNAi was determined at intervals across lifespan (**p* < 0.05 and ****p* < 0.001 versus control RNAi, *n* = 3 worm pellets). **h** Expression levels of *ftn-1* under *ftn-1/C54F6.14* RNAi (#*p* < 0.001 compared to control RNAi, ****p* < 0.001 versus *ftn-1* RNAi, *n* = 3 worm pellets). **i** Quantification of $Fe^{2+}/Fe^{3+}$ iron contents of nematode with *ftn-1/C54F6.14* RNAi at intervals across lifespan (#*p* < 0.001 compared to control RNAi, ***p* < 0.01 and ****p* < 0.001 versus the *ftn-1*

RNAi, *n* = 3 worm pellets). **j** Relative Amplex Red fluorescence in supernatant of worms (#*p* < 0.001 compared to control RNAi, **p* < 0.05 and ***p* < 0.01 versus *ftn-1* RNAi, *n* = 3 worm pellets). **k** Total glutathione (GSH) level was normalized to the GSH level in worms exposed to *tdh/F08F3.4* RNAi or threonine (#*p* < 0.001 compared to control RNAi, **p* < 0.05 and ***p* < 0.01 versus *ftn-1* RNAi, *n* = 3 worm pellets). **l** Levels of the malondialdehyde (MDA), were measured and normalized against the mean of control RNAi-treated worms for independent samples (#*p* < 0.001 compared to control RNAi, ***p* < 0.01 and ****p* < 0.001 versus the *ftn-1* RNAi, *n* = 3 worm pellets). **m**–**o** Effects of *tdh/F08F3.4* RNAi or threonine with ferroptosis inducer erastin or diethyl maleate (DEM) to induce GSH depletion as relative levels of **m** ROS, (**n**) GSH, and **o** MDA with ferroptosis inhibitor, ferrostatin-1 (****p* < 0.001 versus control RNAi, *n* = 3 worm pellets). Overall differences between conditions were analysed by two-way ANOVA. Differences in individual values or between two groups were determined using two-tailed *t*-tests (95% confidence interval). Error bars represent the mean ± s.d. Source data are provided as a Source Data file.

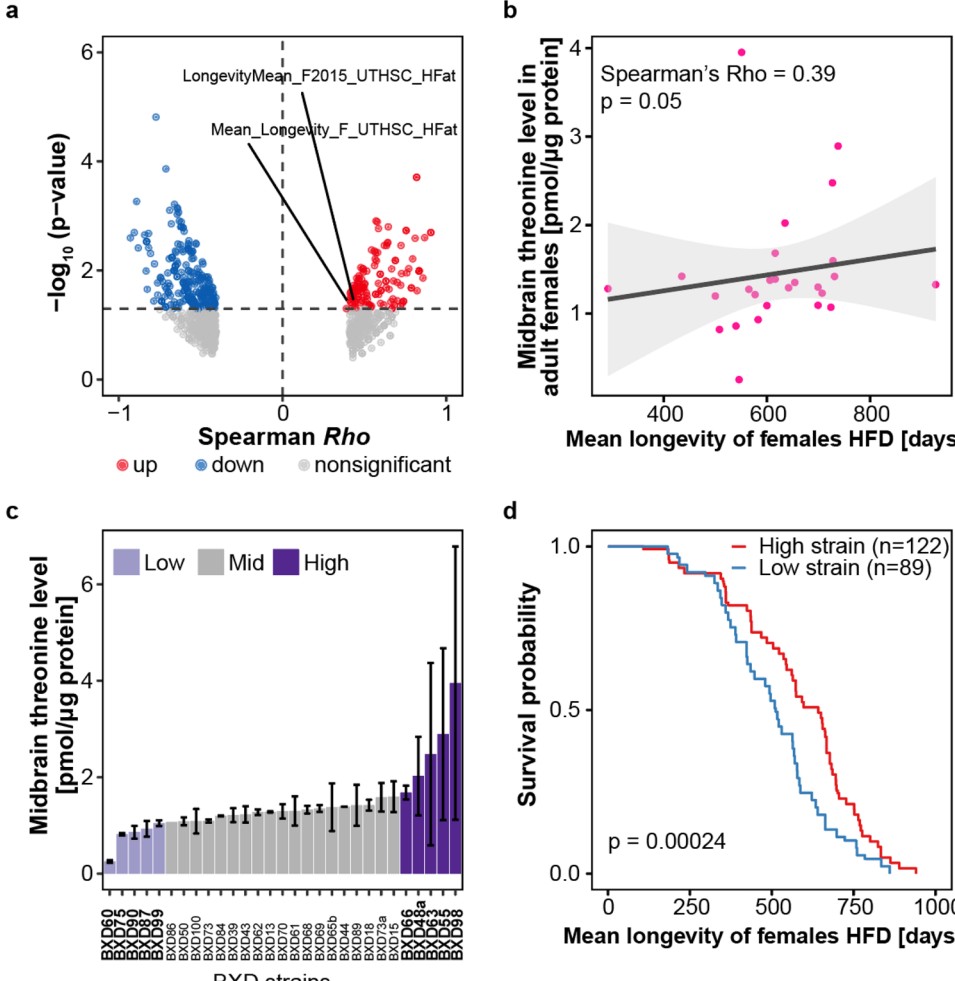

**Fig. 5 | The relation of midbrain threonine concentration of male mice with their lifespan. a** The midbrain threonine levels in male adult male mice with lifespan under control diet (CD) or high fat diet (HFD) condition. **b** The correlation of midbrain threonine level in adult males with mean lifespan of HFD mice (*P* value = 0.05, spearman *Rho* = 0.54). **c** Box plot showing midbrain threonine level across the strains of BXD mouse genetic reference population. **d** Survival probability for BXD strains with high (*n* = 122) or low (*n* = 89) levels of midbrain threonine mice (*P* value = 0.00024). *P* value determined by two-sided log-rank test.

beads, and centrifuged. The supernatant was then filtered through 5-kDa cut-off filters (Ultrafree-MC PLHCC, Merck KGaA, Darmstadt, Germany) to remove macromolecules. The filtrate was centrifugally concentrated and diluted in 25 μl of ultrapure water for CE/TOF-MS analysis. For LC/TOF-MS analysis, frozen worm samples were mixed with 500 μl of 1% formic acid in acetonitrile (v/v) containing internal standards, homogenized, and centrifuged. The supernatant was filtered through a 3-kDa cut-off filter (Nanocep 3 K Omega, Pall Corporation, MI, USA) to remove proteins and then passed over a column (Hybrid SPE phospholipid 55261-U, Supelco, PA, USA) to remove phospholipids. The column eluate was desiccated and suspended in 100 μl of 50% isopropanol. Three independently collected worm pellets were used for each assay with CE-TOF/MS system and 1200 series Rapid Resolution system SL, subsequently LC/MSD-TOF (Agilent Technologies, CA, USA).

### Assay for threonine levels in *C.elegans*
Synchronized adult worms were collected, washed 3 times with M9 buffer, and flash frozen pellets were ground for lysis. Worms were lysed with lysing matrix C tubes (#6912-100, MP Biomedicals, CA, USA) and the TissueLyser II (Qiagen, Hilden, Germany) high speed benchtop homogenizer in the 4 °C room using a setting of 30 Hz for 2 min, then incubated on ice for 1 min, and repeated 3 times. Lysed worms were centrifuged (13,200 rpm, 10 min, 4 °C) to remove debris, and the

supernatant was used. Threonine contents were measured with a threonine assay kit (ab239726, Abcam, MA, USA) to normalization with total protein concentration (BCA assay, Pierce, IL, USA).

### Preparation of threonine supplementary plates
The amino acid threonine was acquired (Sigma-Aldrich, MO, USA) and stock of 1 M was made by dissolving in water. The pH of stock solution was adjusted to 6.0−6.5, the same pH as the vehicle plates, with sodium hydroxide and the threonine stock solution was sterilized with a 0.45 μm Millipore filter (Millipore, MA, USA) before use. Threonine treatment was performed by supplementing the compound in a suspension of heat-inactivated OP50 (65 °C, 30 min with 1 mM magnesium sulfate and 5 mg/ml cholesterol), unless it applied with RNAi clones.

### RNAi in *C. elegans*
For RNAi-mediated gene knockdown, we fed worms HT115 (DE3) expressing target gene double stranded RNA (dsRNA) from the pL4440 vector. The clones for RNAi against genes were obtained from the *C. elegans* ORF-RNAi Library (Thermo Scientific, Horizon Discovery, MA, USA). All clones were verified by sequencing (Macrogen, Seoul, Korea) and efficient knockdown was confirmed by quantitative RT-PCR (qRT-PCR) of the mRNA. The primer sequences used for qRT-PCR are provided in Supplementary Table 2. Bacteria were

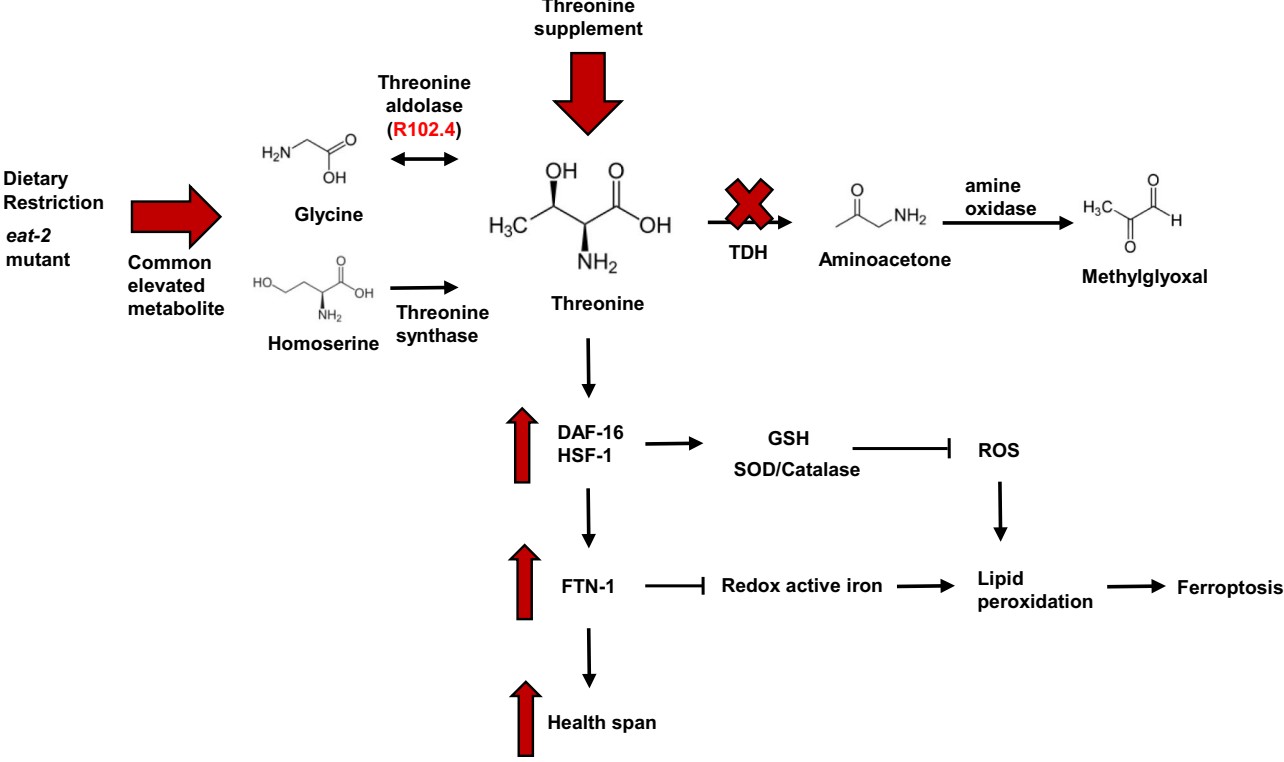

**Fig. 6 | Mechanistic model of threonine-mediated longevity.** Summary of the effects of *tdh/F08F3.4* RNAi or threonine supplementation on DAF-16/HSF-1 and ferroptosis signalling results in healthspan extension. Threonine catabolism impairment by *tdh/F08F3.4* RNAi leads to increased threonine and decreased formation of methylglyoxal. Threonine increased SOD/catalase activity and GSH expression by DAF-16/HSF-1 activation. Subsequently, and dependent on DAF-16 and HSF-1, activation of ferritin protein, FTN-1, reduced redox active iron contents and eventually ferroptosis, which mediates lifespan extension and health parameters.

spotted on NGM plates containing additional 1 mM isoprophyl-β-D-thiogalactoside (IPTG) (Sigma–Alrich, MO, usa). Incubation with dsRNA was initiated 64 h after population synchronization and then young adult worms were tranferred to the respective treatment plates. We confirmed the RNAi knockdown of *tdh* by qRT-PCR and transcript levels of *tdh* were reduced by 71% in larvae that were cultivated on the corresponding dsRNA-expressed bacteria.

### Lifespan analysis
Lifespan assays were carried out at 20 °C on NGM agar plates using standard protocols and were replicated in three independent experiments. *C.elegans* were synchronized with 70 µl M9 buffer, 25 µl bleach (10% sodium hypochlorite solution) and 5 µl of 10 N NaOH. Either a timed 64 h of egg lay or egg preparation, around 150 young adults were transferred to fresh NGM plates containing 49.5 µM 5-fluoro-2′-deoxyuridine (FUDR, Sigma–Aldrich, MO, USA) to prevent progeny production. All compounds were blended into the NGM media after autoclaving and before solidification. Living worms were transferred to assay plates, which were seeded with OP50 or a designated RNAi feeding clone with 50 µg ml$^{-1}$ ampicillin (Sigma-Aldrich, MO, USA) every other day. Worms that crawled off the plates or exhibited internal progeny hatching were censored and those that showed no reaction to gentle stimulation were scored as dead. Lifespan data were analyzed using R software (ver. 4.1.0, "coin" package); Kaplan–Meier survival curves were depicted and *P* values were calculated using the log-rank (Mantel-Cox) test. All lifespan data are described in Supplementary Table 1.

### Triglyceride quantification
Triglyceride (TG) contents were measured using the triglyceride colorimetric assay kit (Abcam, MA, USA) according to the manufacturer's protocol. Briefly, frozen worm pellets were placed in liquid nitrogen with 5% Triton X-100. The pellets were sonicated and diluted for protein determination by BCA assay (Pierce, IL, USA). The samples were heated till 80 °C and shaken for 5 min then, samples were cool-down to room temperature to solubilize all the triglycerides. TG contents were normalized relative to protein contents and three independently collected worm pellets were assayed for each experimental condition.

### Superoxide dismutase and catalase activity assay
To assess antioxidative enzyme activities, 10-day-old worms were ground in liquid nitrogen. Superoxide dismutase (SOD) and catalase activity levels were measured with the SOD colorimetric activity kit (Invitrogen, CA, USA) or Catalase activity colorimetric/fluorometric assay kit (Biovision, CA, USA), respectively. Worm pellet samples were assessed according to manufacturer's protocol in triplicate. SOD and catalase activity were determined using standard curves followed by normalization to protein concentration using the BCA protein assay (Pierce, IL, USA).

### Stress assays
Synchronized worms were treated for each condition for 10 days. For the oxidative stress assay, 20 worms were placed on solid NGM plates containing 0.4 M paraquat (PESTANAL, Sigma-Aldrich, MO, USA) for 3 h. Assays were repeated 9 times and the paraquat plates were freshly prepared on the day of the assay. For the thermotolerance assay, 18–28 worms were exposed to 35 °C for 16 h and the survival of the worms was counted. Assays were performed in triplicate and survival rate was estimated.

### Lipofuscin (age pigment) analysis
Nematodes were synchronized and treated for 10 days with vehicle, control vector or threonine, *tdh*/F08F3.4 dsRNA from L4 larvae stage

(Day 0). Day 10 worms were distributed on a 96-well plate (Porvair Sciences black with glass-bottomed imaging plates, #324002). Lipofuscin auto-fluorescence was determined using a fluorescence plate reader (Synergy H1, BioTek, VT, USA; excitation: 390–410, emission: 460–480) with normalization to the stable signal of the worms (excitation: 280–300, emission: 320–340) as blank. For measuring lipofuscin accumulation, worms were washed with M9 buffer 3 times and fixed for 10 min in 4% paraformaldehyde, washed again and incubated in 0.1% Triton X-100 for 10 min. After washing, the cover slips were mounted onto glass slides and visualized under a confocal laser scanning microscope (LSM8, Carl Zeiss, Jena, Germany) (with excitation at 350 nm and emission at 420 nm). The fluorescence intensities were calculated densitometrically with ZEN Lite software (Carl Zeiss, Jena, Germany) by measuring the average pixel intensity.

### Movement and body size assay

On day 10, 10–15 worms were transferred to fresh NGM plates and 30 s movements were recorded with a microscope system (Olympus SZ61 microscope with Olympus camera eXcope T300, Olympus, Tokyo, Japan). Subsequently, 5 independent movement clips per experimental condition were analyzed with TSView 7 software (ver. 7.1) and average speed was calculated as distance (mm) per second.

For body size measurements, Day 10 worms were washed with M9 buffer 3 times and fixed for 10 min in 4% paraformaldehyde, washed again and incubated in 0.1% Triton X-100 for 10 min. After washing, the cover slips were mounted onto glass slides and visualized using a microscope system (Olympus SZ61 microscope with Olympus camera eXcope T300) and body size was measured with TSView 7 software.

### Fertility assay

Worms were synchronized and single L4 nematodes were transferred on single plates applying control or treatment RNAi bacteria or reagent then moved to fresh plates. Progeny of worms were allowed to hatch and were counted.

### Pharyngeal pumping rates of *C.elegans*

The pharyngeal pumping of 30 worms per each condition were assessed for 1 min using a SZ61 microscope (Olympus, Tokyo, Japan). Pharyngeal contractions were recorded with Olympus camera eXcope T300 at 18-fold optical zoom and videos were played back at 0.5× speed and pharyngeal pumps counted.

### Assay for methylglyoxal (MGO)/Glutathione (GSH) rate in *C. elegans*

For detection of MGO using Methylglyoxal ELISA kit (Abcam, MA, USA), a frozen worm pellet was used and the sample was prepared according to manufacturer's protocol. Glyoxalase activity was also measured with Glyoxalase I colorimetric assay kit (Abcam, MA, USA). Quantification of GSH in worms was measured using the 5,5′-dithio-bis-(2-nitrobenzoic acid) (DTNB) cycling method originally created for cell and tissue samples but optimized for whole worm extracts. Based on a protocol[58], we detected a 5′-thio-2-nitrobenzoic acid (TNB) chromophore, reaction compound of DTNB and GSH, with maximum absorbance of 412 nm using a GSH quantification kit (Dojindo, Kumamoto, Japan).

### DAF-16::GFP / HSF-1::GFP localization assay in *C. elegans*

The *daf-16::gfp* (TJ356) and *hsf-1::gfp* (OG532) strains were used and their fluorescence was visualized using a confocal laser scanning microscope with inverted stand (LSM8, Carl Zeiss, Jena, Germany) (with excitation at 488 nm and emission at 535 nm). For each condition, the nucleus/intermediate/cytoplasm fluorescence or fluorescence intensity of 30–40 worms were detected densitometrically

with ZEN Lite software (Carl Zeiss, Jena, Germany) with triplicate attempts.

### RNA extraction and quantitative real-time PCR (qRT-PCR) in *C. elegans*

Approximately 500 worms were prepared in three biological replicates per each condition at the desired stage. Total RNA was prepared using RNeasy mini kit (Qiagen, Hilden, Germany) then, RNA concentration and quality was measured with a SPECTROstar Nano (BMG Labtech, GmbH, Germany). cDNAs were prepared using SuperScript III reverse transcript kit (ThermoFisher Scientific, MA, USA). TaqMan RT-PCR technology (7500Fast, Applied Biosystems, CA, USA) was used to determine the expression levels of selected target genes with TaqMan site-specific primers and probes (Thermo Scientific, MA, USA). The process included a denaturing step performed at 95 °C for 10 min followed by 50 cycles of 95 °C for 15 s and 60 °C for 1 min. The reactions were performed in triplicate and samples were analyzed by ddCT method with normalization to the reference genes *cdc-42* and *Y45F10D.4* levels[59].

### RNA sequencing (RNA-seq), data analysis, and visualization

RNA was extracted and assigned using Bioanalyzer 2100 (Agilent Technologies, CA, USA) in combination with RNA 6000 nano kit. Matched samples with high RNA integrity number scores proceeded to sequencing. For library preparation an amount of 2 mg of total RNA per sample was processed using Illumina's TruSeq RNA Sample Prep Kit (Illumina, CA, USA) following the manufacturer's instruction. Quality and quantity of the libraries was determined using Agilent Bioanalyzer 2100 in combination with FastQC v0.11.7, Sequencing were done on a HiSeq4000 in SR/50 bp/high output mode at the Macrogen Bioinformatics Center (Macrogen, Seoul, Korea).

Libraries were multiplexed five per lane. Sequencing resulted in 35 million reads per sample. Sequence data were extracted in FastQ format using bcl2fastq v1.8.4 (Illumina) and used for mapping. FASTQ output files were aligned to the WBcel235 (February 2014) *C. elegans* reference genome using STAR[60]. These files have been deposited at the Sequence Read Archive (SRA) with the accession number SRP341842. Samples averaged 75% mapping of sequence reads to the reference genome. We filtered out transcripts with Trimmomatic 0.38 platform[61]. Differential expression analysis was performed using HISAT2 version 2.1.0, Bowtie2 2.3.4.1, with StringTie version 2.1.3b[62,63]. Intersection of differentially expressed genes (DEGs) of *tdh/F08F3.4* RNAi (padj < 0.05 in edgeR, DEseq) was performed with data from indicated published datasets by Venn analysis using the ggVennDiagram package in R software. Human ortholog matching was performed using WormBase, Ensembl, and OrthoList2[64]. Gene lists were evaluated for functional classification and statistical overrepresentation with Database for Annotation, Visualization, and Integrated Discovery (DAVID) version 6.8 and Kyoto Encyclopedia of Genes and Genomes (KEGG) pathway database.

### Iron assay

Iron contents were measured using the Iron assay kit (Sigma–Aldrich, MO, USA) according to the manufacturer's protocol. Briefly, worm pellets were frozen in liquid nitrogen with 5% Triton X-100. The pellets were sonicated with 5 volumes of iron assay buffer and centrifuged at 13,000 g for 10 min at 4 °C to remove insoluble material. Released iron into the acidic buffer was reacted with a chromagen resulting in a colorimetric (593 nm) product, proportional to the iron present. Ferrous/Ferric iron was determined using standard curves followed by normalization to protein concentration using the BCA protein assay (Pierce, IL, USA). Three biological replicates of the assay were performed.

## Quantification of reactive oxygen species level

On day 10, 5 plates of worms were washed with M9 buffer and incubated with 100 µM Amplex Red probe (Invitrogen, CA, USA) in Krebs-Ringer Phosphate buffer (145 mM NaCl, 5.7 mM Na$_2$PO$_4$, 4.86 mM KCl, 0.54 mM CaCl$_2$, 1.22 mM MgSO$_4$ and 5.5 mM glucose, pH 7.4) containing 0.2 U ml$^{-1}$ of horseradish peroxidase. After occasional mixing in the dark for 3 h, fluorescence intensity was determined using a fluorescence plate reader (Synergy H1, BioTek, VT, USA; excitation: 571, emission: 585) by normalization using protein input with BCA protein assay (Pierce, IL, USA). Three biological replicates of the measurement were performed.

## Lipid peroxidation assay

For measurement of lipid peroxidation level, we performed a Thiobarbituric acid reactive substances (TBARS) assy kit (Cayman Chemical, MI, USA) to detect malondialdehyde (MDA) level, according to manufacturer's instructions. Replicate experimental samples were collected and frozen worms were homogenized by TissueLyser II (Qiagen, Hilden, Germany) in the 4 °C room using a setting of 30 Hz for 2 min, then incubated on ice for 1 min, and repeated 3 times. Lysed worms were centrifuged (13,200 rpm, 15 min, 4 °C) to remove debris, and the supernatant retained. The protein concentration was determined with BCA protein assay (Pierce, IL, USA) and equivalent 20–25 µg of total protein was used for the assay. For conditions with acute glutathione depletion, Day 2 animals were treated with 20 mM diethyl maleate (DEM) or 5 mM of erastin for 6 h prior to collection. Three biological replicates of the assay were performed.

## Statistical analysis

All experiments were repeated at least three times with identical or similar results. Data represents biological replicates. Adequate statistical analysis was used for every assay. Data meet the hypothesis of the statistical tests described for each experiments. Data are expressed as the mean ± s.d. in all figures unless stated otherwise. R software (ver. 4.1.0) and Excel 2016 (Microsoft, NM, USA) were used for statistical analyses. The $p$-values $*p < 0.05$, $**p < 0.01$, $***p < 0.001$, and $^{\#}p < 0.01$ were considered statistically significant.

## Reporting summary

Further information on research design is available in the Nature Research Reporting Summary linked to this article.

## Data availability

There is no restriction on data availability. The RNA sequence data used in this study have been deposited in NCBI's SRA and are accessible through submission number PRJNA772096. Source data for all the individual $P$ values are provided with this paper. All data for the BXD mouse reference population can be found on the GeneNetwork website [http://www.genenetwork.org/). Source data are provided with this paper.

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

## Acknowledgements

*C. elegans* strains used in this research were provided by the *Caenorhabditis* Genetics Center (University of Minnesota, USA), which is funded by the NIH Office of Research Infrastructure Programs. D.R. was supported by grants from the National Research Foundation of Korea (NRF) funded by the Korean government (MSIT, 2020R1A2C2010964 and 2021R1A5A8029876).

## Author contributions

J.K. created the experimental design and performed all experiments, except as stated otherwise. Y.J. contributed to the analysis of metabolomics, RNA sequencing data, and visualization. D.C. performed experiments on isolated nematodes. J.K. and D.R. designed the study and wrote the manuscript; D.R. edited the manuscript; and J.K. and D.R. conceived and designed the research, analysed and interpreted the data, and administered the project. All authors discussed the results and commented on the manuscript.

## Competing interests

The authors declare no competing interests.
