## [Peer Review File · Nature Communications]

L-threonine promotes healthspan by expediting
ferritin-dependent ferroptosis inhibition in *C. elegans*REVIEWER COMMENTS

Reviewer #1 (Remarks to the Author):

The authors found that threonine levels increased with lifespan-extending dietary restriction. Since it is known that threonine supplementation increases lifespan, the authors determined the molecular mechanism and found that threonine increases the activity of FAF-16 and HSF-1 transcription factors that upregulated the expression of ferritin (FTN-1) that bound iron to decrease ferroptosis to extend lifespan. The research is strong and the paper is well-written. But there are many changes that need to be made before the manuscript is acceptable for publication.

Comment 1: A) Figure 2A and Figure 5: To clarify, can you write threonine aldolase in black font next to R102.4? It would also be helpful to write that the other product of threonine catabolism by R102.4 is acetaldehyde since there is room in the top left corner. Acetaldehyde levels may limit the conversion of glycine to threonine.

B) Results section: I wouldn't call R102.4 (threonine aldolase) a threonine synthesizing (or "threonine-elaborating" (line 91)) enzyme. Even though it is reversible, it likely favors threonine catabolism under many conditions. I would call it a threonine-metabolizing enzyme.

C) Figure 2A: α -ketobutylate \rightarrow α -ketobutyrate

Figure 2A: It is not clear why there is an arrow from threonine to pyruvate. Shouldn't the arrow be from glycine to pyruvate, as glycine is converted to serine and then to pyruvate? Also, no arrow should be placed from pyruvate to valine and leucine as they are essential amino acids and cannot be synthesized from pyruvate. And no arrow should be shown from 2-oxobutanoate to isoleucine as isoleucine is also an essential amino acid.

D) I could not find evidence that *C. elegans* contains a threonine synthase gene to synthesize threonine from homoserine. Please delete this connection from the figure 2A unless this gene exists.

E) I also do not see the justification for the arrow from valine and leucine to threonine deaminase. So, major revisions are needed on Figure 2A.

F) In the legend for Figure 2C indicate which day of the lifespan the metabolite measurements were made.

In the legend for Figure 2E-G indicate which day of the lifespan the protein abundance measurements were made.

Comment 2: Please reference the original report of threonine extending lifespan in *C. elegans*. Edwards, C. et al. BMC Genet. 2015 Feb 3;16(1):8. In that report they saw a lifespan extension at 10 mM threonine, but not at 1 mM threonine. It would be interesting to determine the effects of 10 mM and 20 mM threonine on lifespan. Perhaps threonine can extend lifespan at high and low (0.2 mM) concentrations by different mechanisms.

Comment 3: Can you mention in the discussion that the human threonine aldolase and threonine dehydrogenase genes have mutated so that functional proteins are no longer expressed? Therefore, threonine deaminase (dehydratase) is the default pathway for human threonine catabolism. Therefore, it may be difficult to apply studies of threonine metabolism in model organisms that have multiple pathways for threonine catabolism to humans that just have one. However, mice have both functional threonine aldolase and threonine dehydrogenase.

Comment 4: It is logical that F08F3.4/tdh knockdown to increase threonine level increased the lifespan of *daf-2* mutants as *daf-2* mutants have been described to have lowered threonine levels. Please indicate this and reference the paper. Castro, C. et al. Mol Biosyst. 2013 Jul 1; 9(7): 1632–1642. However, another report found no change in threonine level in the *daf-2* mutant. Martin, F. et al. J. Proteome Res. 2011, 10, 3, 990–1003.

Comment 5: It is unclear why the authors use the term siRNA. In *C. elegans* research the term RNAi is almost always used. Using siRNA is thus misleading.

Comment 6: Figure 4J and 4M: Amplex Red specifically measures hydrogen peroxide. So, the Y-axis should be labeled Relative hydrogen peroxide level.

Comment 7: In Figure 4H-L it indicates that the authors are using FTN-1 RNAi (feeding *ftn-1* bacterial RNAi clone). It would be more appropriate to use the *C. elegans* *ftn-1* mutant strain like was done for the lifespan experiments in Figure 3. Feeding two RNAi clones together such as *ftn-1* and *tdh* likely dilute the effect of each one by half. Therefore, controls would need to be performed where each RNAi clone is diluted 50% with an empty vector RNAi clone. Please explain how the experiment was performed in more detail. If the experiment was performed with the *ftn-1* mutant strain please change the graph accordingly. If not, it would likely be best to perform the experiments again but this time using the *ftn-1* mutant strain.

Comment 8: In Figure 5 there is an arrow from ROS to amine oxidase. Please explain the meaning of this in the legend and provide a reference if ROS increases the expression of amine oxidase in *C. elegans* or cells from another organism.

Comment 9: Research from R. Houtkooper's lab should be discussed. They found that during *C. elegans* aging the levels of almost all amino acids decreased, with the exception of glycine and aspartate that slightly increased, with glycine starting to increase on day 9 (Gao, A. *Sci Rep.* 2017; 7: 2408). With DR this glycine can be converted to threonine. They also performed metabolomics experiments identified 17 out of the 20 amino acids (threonine was one of the 3 amino acids not identified) and found that the vast majority (including glycine) were decreased in long-lived *eat-2* and *daf-2* mutants. Gao, A et al. *Exp Gerontol.* 2018 Nov; 113: 128–140. They found R102.4 to decrease from day 1 to day 4 and then restored to day 1 level on day 9 (Liu, Y. et al. *PLoS Genet.* 2019 Mar; 15(3): e1007633.).

Comment 10: Line 145: Please explain how ferrous iron is required for ferroptosis and the significance of the redox-active iron ratio (Fe^{2+}/Fe^{3+}) (perhaps in the Discussion).

Comment 11: Line 170: You mention lipofuscin: Blue autofluorescence that increases with aging in *C. elegans* has show to be due to release of lysosomal anthranilate esters into the cytoplasm. It is unclear if *C. elegans* accumulates lipofuscin with aging. Coburn, C. and Gems D. *Front Genet.* 2013 Aug 7;4:151.doi: 10.3389/fgene.2013.00151.

Comment 12: Line 182: specified -> measured

Comment 13: Line 208: [typo] level. and blocks the generation of. -> level, and blocks the generation of ROS.

Comment 14: Line 219: mammals and humans -> mammals including humans

Comment 15: Line 496: 96well -> 96-well

Reviewer #2 (Remarks to the Author):

The manuscript is original, being significant for studies on longevity and cell survival mechanisms. The work is carefully performed, the methodology is well described, and the conclusions are supported by the experiments.

Specific points to address:

- How the DR-associated metabolites from *C. elegans* compare to the human DR-associated metabolites? The authors should include these data.

- The threonine effect seems associated with a narrow concentration range. How would this range be applicable to humans or other species?

- Is ferrostatin administration also increasing longevity?

- what is the effect of ferrostatin in *ftn-1/C54F6.14* RNAi worms?

Reviewer #3 (Remarks to the Author):

Main comments:

1. The authors assert that threonine level is the primary driver for the increased lifespan observed in the study, rather than its metabolites or precursors (line 97). This is not well supported by evidence and data presented. What appears to be clear is that glutathione and malondialdehyde levels are being altered as a result of threonine treatment or enzyme knockdowns, and these are both relatively well-established molecular markers of oxidative stress response and healthspan in model organisms. It is possible that threonine treatment or enzymatic knockdowns could lead to wholesale changes in the metabolome that triggered transcriptional and signalling regulations etc. The authors could have performed follow-up metabolomics analysis to address this by directly comparing the impact of threonine supplementation on the metabolome (for example by comparing worms fed with 200µm threonine to worms fed with no/very little threonine under dietary restriction).

2. The study data clearly show that the impact of threonine supplementation on lifespan in *C.elegans* is reversed at high concentration. The impact of threonine supplementation on lifespan appears to be

tightly regulated/ limited by other additional biological processes. Could the authors have explored this further?

3. Threonine pathway is closely related to other amino acid pathways (eg. glycine) in *C.elegans*, and threonine maybe interconverted to and from other metabolites. Is the mechanism underlying lifespan enhancement likely to be same for glycine and threonine supplementations? Could glycine supplementation also trigger ferritin response?

4. The authors have argued for additional studies to evaluate whether the threonine effect on longevity translate to mammals and humans. The authors may want to add to the discussion the threonine genes they investigated that are present in humans.

Minor comments:

5. Fig. 1a - the metabolite class labelling is problematic. It appears that they are supposed to be labelled by pathway/ compound class. However, this does not appear to be the case, at least for the metabolites grouped under the "fatty acid metabolism" labels.

6. Ferritin expression has been shown to be regulated by inflammatory status, were IL-6 /TNF-alpha levels altered as a result of threonine treatment or enzyme knockdowns?

7. mTOR and AMPK signalling are key longevity pathways, does threonine treatment alters mTOR and AMPK activity?

8. It would appear that the metabolomics analysis was performed by Human Metabolome Technologies. I know they offer multiple different platform services; did you perform the global profiling assay or the targeted assay for this study? I think you need to specify in the supplementary data file which of the metabolites were measured using CE-MS, and which of those were measured using LC-MS, and LC-MS/MS. I suspect most of the metabolites were not measured using LC-MS/MS, and if that's the case, the abstract should be corrected.

Reviewer #1:

The authors found that threonine levels increased with lifespan-extending dietary restriction. Since it is known that threonine supplementation increases lifespan, the authors determined the molecular mechanism and found that threonine increases the activity of FAF-16 and HSF-1 transcription factors that upregulated the expression of ferritin (FTN-1) that bound iron to decrease ferroptosis to extend lifespan. The research is strong and the paper is well-written. But there are many changes that need to be made before the manuscript is acceptable for publication.

Dear Reviewer #1,

First of all, we would like to thank you very much for your constructive and insightful comments. Below, in blue font, we have summarised our opinion and what we have done further, based on your comments.

Comment 1:

A) Figure 2A and Figure 5: To clarify, can you write threonine aldolase in black font next to R102.4? It would also be helpful to write that the other product of threonine catabolism by R102.4 is acetaldehyde since there is room in the top left corner. Acetaldehyde levels may limit the conversion of glycine to threonine.

A) To clarify, we changed "R102.4" to "Threonine aldolase (R102.4)" in Figures 2a and 6. We also included "acetaldehyde." as a product of threonine catabolism by threonine aldolase in Figure 2a.

B) Results section: I wouldn't call R102.4 (threonine aldolase) a threonine synthesizing (or "threonine-elaborating" (line 91)) enzyme. Even though it is reversible, it likely favors threonine catabolism under many conditions. I would call it a threonine-metabolizing enzyme.

B) In the results section, we corrected the paragraph in R102.4 to "threonine metabolising enzyme" in lines 72, 75, 87, 195, and 209.

C) Figure 2A: α -ketobutylate \rightarrow α -ketobutyrate

Figure 2A: It is not clear why there is an arrow from threonine to pyruvate. Shouldn't the arrow be from glycine to pyruvate, as glycine is converted to serine and then to pyruvate? Also, no arrow should be placed from pyruvate to valine and leucine as they are essential amino acids and

cannot be synthesized from pyruvate. And no arrow should be shown from 2-oxobutanoate to isoleucine as isoleucine is also an essential amino acid.

C) In Figure 2a, α -ketobutylate has been replaced with α -ketobutyrate

We corrected threonine metabolism in Figure 2a. The arrows pointing to three essential amino acids—valine, leucine, and isoleucine—were removed. To avoid any confusion, we also excluded pyruvate.

D) I could not find evidence that *C. elegans* contains a threonine synthase gene to synthesize threonine from homoserine. Please delete this connection from the figure 2A unless this gene exists.

D) In Figure 2a, we eliminated the connection from homoserine to threonine synthesis.

E) I also do not see the justification for the arrow from valine and leucine to threonine deaminase. So, major revisions are needed on Figure 2A.

E) In Figure 2a, we also deleted the connection between valine and leucine to threonine deaminase.

F) In the legend for Figure 2C indicate which day of the lifespan the metabolite measurements were made. In the legend for Figure 2E-G indicate which day of the lifespan the protein abundance measurements were made.

F) In the legend for Figure 2, we indicated the age (day) of worms, which was used in metabolite measurement and protein abundance.

Comment 2: Please reference the original report of threonine extending lifespan in *C. elegans*. Edwards, C. et al. *BMC Genet.* 2015 Feb 3;16(1):8. In that report they saw a lifespan extension at 10 mM threonine, but not at 1 mM threonine. It would be interesting to determine the effects of 10 mM and 20 mM threonine on lifespan. Perhaps threonine can extend lifespan at high and low (0.2 mM) concentrations by different mechanisms.

The report, "*Edwards, C. et al. BMC Genet. 2015 (PMID: 25643626)*" was cited as reference No. 10. In that report, they observed a lifespan extension with an increasing concentration of threonine in 1 mM, 5 mM, and significantly in 10 mM. We performed the lifespan assay with 10, 20, and 40 mM of threonine. The threonine supplementation increased lifespan at both low (100 to 400 μ M, Figures 1c and d) and high (10 and 20 mM, Extended Data Fig. 1h-m) concentrations, except at 40 mM of threonine supplementation.

Because a high dose of drug treatment frequently causes longevity in *C. elegans* by inhibiting bacterial growth, which mimics the effect of calorie restriction as acts as an artefact, we evaluated the growth rate of *E. coli* (OP50). A high concentration of threonine, as shown in Extended Data Fig. 1k-m, reduced the growth rate of *E. coli*, which may have contributed to lifespan extension, at least in part. However, supplementing with 40 mM threonine reduced lifespan, which we presume the chemical toxicity results in a reduced lifespan of worms. Furthermore, we believe that high concentrations of threonine, greater than 10 mM, are difficult to find in nature. It is most likely only found in experimental conditions, such as genetically engineered *E. coli* (PMID:33653345). As a result, the low concentrations investigated in this report (100-400 μ M) are reasonable and could be effective interventions. We added these results in Extended Data Fig. 1h-m and Extended Table 1.

Comment 3: Can you mention in the discussion that the human threonine aldolase and threonine dehydrogenase genes have mutated so that functional proteins are no longer expressed? Therefore, threonine deaminase (dehydratase) is the default pathway for human threonine catabolism. Therefore, it may be difficult to apply studies of threonine metabolism in model organisms that have multiple pathways for threonine catabolism to humans that just have one. However, mice have both functional threonine aldolase and threonine dehydrogenase.

The reviewer mentioned threonine dehydratase as the default pathway for human threonine catabolism. As a result, applying this research to model organisms with multiple pathways for threonine catabolism to humans with only one is difficult. However, mice have both functional threonine aldolase and threonine dehydrogenase (PMID: 15757516), and we discuss the role of threonine in mouse longevity.

Comment 4: It is logical that F08F3.4/tdh knockdown to increase threonine level increased the lifespan of *daf-2* mutants as *daf-2* mutants have been described to have lowered threonine levels. Please indicate this and reference the paper. Castro, C. et al. Mol Biosyst. 2013 Jul 1; 9(7): 1632–1642. However, another report found no change in threonine level in the *daf-2* mutant. Martin, F. et al. J. Proteome Res. 2011, 10, 3, 990–1003.

We observed further lifespan extension in *daf-2* mutants with F08F3.4/tdh knockdown to increase threonine levels (Figure 3e and Extended Data Fig. 4f-g). It makes sense because *daf-2* mutants have been reported to have lower threonine levels. As reference No. 31, we cited the paper "Castro, C. et al. Mol Biosyst. 2013 (PMID: 25762526)."

Comment 5: It is unclear why the authors use the term siRNA. In *C. elegans* research the term RNAi is almost always used. Using siRNA is thus misleading.

Regarding the reviewer's comment, we changed the term "siRNA" to "RNAi" throughout the manuscript and in Figure 2m-r to avoid any confusion.

Comment 6: Figure 4J and 4M: Amplex Red specifically measures hydrogen peroxide. So, the Y-axis should be labeled Relative hydrogen peroxide level.

In Figures 4j and m, we also corrected the Y-axis label to "Relative hydrogen peroxide level".

Comment 7: In Figure 4H-L it indicates that the authors are using FTN-1 RNAi (feeding *ftn-1* bacterial RNAi clone). It would be more appropriate to use the *C. elegans* *ftn-1* mutant strain like was done for the lifespan experiments in Figure 3. Feeding two RNAi clones together such as *ftn-1* and *tdh* likely dilute the effect of each one by half. Therefore, controls would need to be performed where each RNAi clone is diluted 50% with an empty vector RNAi clone. Please explain how the experiment was performed in more detail. If the experiment was performed with the *ftn-1* mutant strain please change the graph accordingly. If not, it would likely be best to perform the experiments again but this time using the *ftn-1* mutant strain.

We apologise for any confusion. In fact, Figures 4h-l show the experiment with the *ftn-1* mutant strain with *tdh* RNAi. Thus, we changed the label to "*ftn-1(ok3625)* + *tdh* RNAi."

Comment 8: In Figure 5 there is an arrow from ROS to amine oxidase. Please explain the meaning of this in the legend and provide a reference if ROS increases the expression of amine oxidase in *C. elegans* or cells from another organism.

In Figure 6, we meant that ROS accelerates the conversion of amine oxidase activity for the conversion of aminoacetone to MGO. To avoid misunderstanding, we deleted the arrow from ROS to amine oxidase.

Comment 9: Research from R. Houtkooper's lab should be discussed. They found that during *C. elegans* aging the levels of almost all amino acids decreased, with the exception of glycine and aspartate that slightly increased, with glycine starting to increase on day 9 (Gao, A. Sci Rep. 2017; 7: 2408). With DR this glycine can be converted to threonine. They also performed metabolomics experiments identified 17 out of the 20 amino acids (threonine was one of the 3 amino acids not identified) and found that the vast majority (including glycine) were decreased in long-lived *eat-2* and *daf-2* mutants. Gao, A et al. Exp Gerontol. 2018 Nov; 113: 128–140. They found R102.4 to decrease from day 1 to day 4 and then restored to day 1 level on day 9 (Liu, Y. et al. PLoS Genet. 2019 Mar; 15(3): e1007633.).

In the discussion, we discussed the study about amino acid levels with *C. elegans* ageing, with glycine starting to increase on day 9 (PMID: 28546536) as Reference No. 46. In addition, we discussed that the "R102.4" level decreased from day 1 to day 4 and was restored to day 1 level on day 9 as Reference No. 7 (PMID: 30300667). In the current study, we examined the metabolomics study on day 10 and a similar tendency was observed.

Comment 10: Line 145: Please explain how ferrous iron is required for ferroptosis and the significance of the redox-active iron ratio (Fe²⁺/Fe³⁺) (perhaps in the Discussion).

At lines 216-220, we discussed the relationship of ferrous iron and the significance of redox-active iron ratio with ferroptosis with reference to 50–51.

Comment 11: Line 170: You mention lipofuscin: Blue autofluorescence that increases with aging in *C. elegans* has show to be due to release of lysosomal anthranilate esters into the cytoplasm. It is unclear if *C. elegans* accumulates lipofuscin with aging. Coburn, C. and Gems D. Front Genet. 2013 Aug 7;4:151.doi: 10.3389/fgene.2013.00151.

To agree with the reviewer's comment and avoid confusion, we deleted the lipofuscin results from Extended Data Fig. 8j-8k and lines 178–180.

Comment 12: Line 182: specified -> measured

At line 166, "specified" was corrected to "measured".

Comment 13: Line 208: [typo] level. and blocks the generation of. -> level, and blocks the generation of ROS.

At line 220, we corrected the omitted typo to "level, and blocks the generation of ROS".

Comment 14: Line 219: mammals and humans -> mammals including humans

At line 231: "mammals and humans" is corrected to "mammals including humans".

Comment 15: Line 496: 96well -> 96-well

At line 403, we corrected "96well" to "96-well".

Reviewer #2:

The manuscript is original, being significant for studies on longevity and cell survival mechanisms. The work is carefully performed, the methodology is well described, and the conclusions are supported by the experiments.

First and foremost, we would like to thank you for your constructive and thoughtful comments. Focusing on your comments, we have summarised our thoughts and actions in blue font below.

Specific points to address:

- How the DR-associated metabolites from *C. elegans* compare to the human DR-associated metabolites? The authors should include these data.

1. We include the mice data about the relation of threonine concentration in the midbrain of male mice with their lifespan in Figure 5. Although we did not contain data about human metabolites, mice have similar functional threonine catabolism pathways and could give a meaningful clue to understanding the impact of threonine in age-associated amino acid metabolism.

- The threonine effect seems associated with a narrow concentration range. How would this range be applicable to humans or other species?

2. To clarify the longevity effects of threonine with a narrow concentration range, we performed the lifespan assay with higher concentrations of 10, 20, and 40 mM. Moreover, we detected that 10–40 mM of threonine inhibited the growth of *E. coli* (OP50), a food source for *C. elegans* (Extended Data Fig. 1k-m).

Reduced OP50 concentration affects *C. elegans* longevity as calorie restriction-like effects at high concentrations of 10 and 20 mM, but chemical toxicity of threonine results in a reduced lifespan at a higher concentration of 40 mM.

Hence, we assume that high concentrations of threonine, like more than 10 mM, are hard to find in nature except in experimental conditions, such as genetically engineered *E. coli* (PMID: 33653345). Hence, it is adequate that the concentration examined in the current study (100–400 μ M) is applicable to other species and may also be applicable to humans.

We supplemented these results in Extended Data Fig. 1h–1m and Extended Table 1.

- Is ferrostatin administration also increasing longevity?

3. Because ferroptosis affects *C. elegans* frailty (Ref. 34) and activity (Ref. 33), we speculated that ferroptosis inhibition by ferrostatin could induce longevity. We performed an additional lifespan assay with ferrostatin administration with a 1, 10, and 100 μ M dose range. As expected, ferrostatin treatment extended the lifespan of *C. elegans* in a dose-dependent manner (Extended Data Fig. 7l-7n). It may be because of the specific alleviating effect of ferrostatin against age-related ferroptosis.

We supplemented these results in Extended Data Fig. 7l–7n and Extended Table 1.

- what is the effect of ferrostatin in *ftn-1/C54F6.14* RNAi worms?

4. To investigate the effect of ferrostatin, we further investigated ROS, GSH, and also MDA levels with *ftn-1/C54F6.14* RNAi worms. As a result, *ftn-1* RNAi greatly increased hydrogen peroxide and also MDA, or reduced GSH levels. However, ferrostatin treatment could not recover these effects (Extended Data Fig. 7r-7t). We thought that ferrostatin-1 effects of ferroptosis inhibition were absolutely associated with ferritin activity. Correspondingly, the lifespan extension effects of ferrostatin-1 were completely diminished in *ftn-1* RNAi animals. These results suggest that the longevity effects of ferrostatin are largely induced by its ferroptosis inhibition capacity.

We added these results in Extended Data Fig. 7o–t and Extended Table 1.

Reviewer #3:

Main comments:

Above all, we appreciate your thoughtful and constructive criticism. Based on your comments, we have compiled our ideas, responses, and reactions in blue font below.

1. The authors assert that threonine level is the primary driver for the increased lifespan observed in the study, rather than its metabolites or precursors (line 97). This is not well supported by evidence and data presented. What appears to be clear is that glutathione and malondialdehyde levels are being altered as a result of threonine treatment or enzyme knockdowns, and these are both relatively well-established molecular markers of oxidative stress response and healthspan in model organisms. It is possible that threonine treatment or enzymatic knockdowns could lead to wholesale changes in the metabolome that triggered transcriptional and signalling regulations etc. The authors could have performed follow-up metabolomics analysis to address this by directly comparing the impact of threonine supplementation on the metabolome (for example by comparing worms fed with 200um threonine to worms fed with no/very little threonine under dietary restriction).

1. We totally agree with Reviewer's comments that glutathione and MDA levels are being altered as a result of threonine treatment or TDH knockdown, and these factors are well-established molecular markers of ferroptosis and healthspan. We corrected the assertive expression "threonine itself" in line 105 and discussed its results, such as GSH, MDA, and FTN-1 levels by transcription factor DAF-16 in the discussion.

2. The study data clearly show that the impact of threonine supplementation on lifespan in *C.elegans* is reversed at high concentration. The impact of threonine supplementation on lifespan appears to be tightly regulated/ limited by other additional biological processes. Could the authors have explored this further?

2. To explain the impact of threonine supplementation with a tightly regulated dose range, we performed additional experiments with higher doses of 10, 20, and 40 mM of threonine on the lifespan. We confirmed that a high dose of threonine (10, 20, and 40 mM) inhibited the *E. coli* OP50 growth and this could affect the lifespan changes of *C. elegans*. Because of the chemical toxicity of high doses of threonine (40 mM), the longevity effects of threonine intervention are tightly regulated and limited by calorie restriction-like biological processes. As a result, we speculate that threonine concentrations greater than 10 mM are unlikely to be found in nature except in experimental conditions, such as genetically engineered *E. coli* (PMID: 33653345), whereas the appropriate dose range of threonine in

this study (100-400 μ M) is capable of extending healthspan in other species.

We inserted these results in Extended Data Fig. 1h-1m and Extended Table 1.

3. Threonine pathway is closely related to other amino acid pathways (eg. glycine) in *C.elegans*, and threonine maybe interconverted to and from other metabolites. Is the mechanism underlying lifespan enhancement likely to be same for glycine and threonine supplementations? Could glycine supplementation also trigger ferritin response?

3. It is reported that the levels of almost all amino acids decreased, with the exception of glycine and aspartase, which slightly increased, with glycine starting to increase on day 9 during *C. elegans* ageing (Ref. 46). And glycine promotes *C. elegans* longevity in a methionine cycle-dependent manner (Ref. 7). We think the possibility that this glycine can be converted to threonine with DR condition in accordance with increased R102.4 expression was restored to day 1 level on day 9. On the other hand, as the reviewer said, to clarify that glycine supplementation could also trigger ferritin-mediated longevity, we performed the following experiments with 50 and 500 μ M of glycine, which were effective in nematode lifespan extension (Ref. 7). As a result, glycine administration could not affect the *ftn-1* expression with aging, Fe²⁺/Fe³⁺ ratio, and MDA levels, although it slightly reduced hydrogen peroxide and increased glutathione levels under ferroptosis-induced conditions. These results suggest that ferroptosis inhibition via FTN-1 activation is specific for threonine supplementation. On the other hand, glycine affects the lifespan of *C. elegans* by different mechanisms with ferroptosis, which could reduce ROS.

We addressed these results in Extended Data Fig. 8o-s.

4. The authors have argued for additional studies to evaluate whether the threonine effect on longevity translate to mammals and humans. The authors may want to add to the discussion the threonine genes they investigated that are present in humans.

4. It is important to evaluate whether the threonine effect on longevity translates to mammals and humans. Human threonine aldolase and threonine dehydrogenase genes have mutated so that functional proteins are no longer expressed. Therefore, threonine deaminase (dehydratase) is the default pathway for human threonine catabolism. Insofar as it may be difficult to apply current studies of threonine metabolism in model organisms that have multiple pathways for threonine catabolism to humans, who just have one. However, mice have both functional threonine aldolase and threonine dehydrogenase. We discussed this and supplemented the mice data of midbrain threonine concentration with male murine lifespan

in Figure 5. Although we lacked human metabolite data, the mice data could help us understand the role of threonine in age-related changes in higher organisms.

Minor comments:

5. Fig. 1a - the metabolite class labelling is problematic. It appears that they are supposed to be labelled by pathway/ compound class. However, this does not appear to be the case, at least for the metabolites grouped under the “fatty acid metabolism” labels.

5. In Figure 1a, we corrected the label to metabolic pathway/compound class. "Fatty acid metabolism" was divided into "Lys degradation" and "Purine metabolism".

6. Ferritin expression has been shown to be regulated by inflammatory status, were IL-6 /TNF-alpha levels altered as a result of threonine treatment or enzyme knockdowns?

6. We checked and confirmed that the RNAseq data of TDH RNAi versus Control RNAi worms showed TNF receptor-associated factor (TRAF) family genes lowered in the threonine catabolism enzyme knockdown condition. TRAF genes are conserved pathways linking cell surface receptors to activation of JNK, ERK, and NF-kappaB (PMID: 29130360). Moreover, we confirmed the pro-inflammatory cytokines of *C. elegans*, pathogen-dependent cytokine-like molecules, and C-type lectin domain-containing proteins (CLECs) have shown decreased levels, and *rde-1* and *rde-4*, which act as signal transducers against viral infections, were also shown to be reduced by TDH RNAi condition. These genes' expression could be checked in the raw data of RNAseq, which has been deposited to the public NCBI SRA database as "PRJNA 772096". Because ferritin expression has been shown to be regulated by inflammatory status, it is reasonable to assume that increasing threonine alters inflammatory response and results in ferritin expression changes.

7. mTOR and AMPK signalling are key longevity pathways, does threonine treatment alters mTOR and AMPK activity?

7. Besides the insulin-like pathway, mTOR and AMPK signalling are also key longevity pathways. In a recent study about mTORC activation by various amino acids, it has been reported that threonine treatment slightly activates mTORC1 (PMID: 32799865). However, in a similar threonine elevation study, mTOR- or AMPK-related genes did not change (Ref.21) and also in our current study (RNAseq result). In threonine treatment, longevity signalling is mainly mediated by HSF-1, DAF-16, and SKN-1 (Ref.21 and current study).

8. It would appear that the metabolomics analysis was performed by Human Metabolome Technologies. I know they offer multiple different platform services; did you perform the global profiling assay or the targeted assay for this study? I think you need to specify in the supplementary data file which of the metabolites were measured using CE-MS, and which of those were measured using LC-MS, and LC-MS/MS. I suspect most of the metabolites were not measured using LC-MS/MS, and if that's the case, the abstract should be corrected.

8. We performed a global profiling assay by Human Metabolome Technologies. We specified and corrected the abstract and method section for assay protocols. We measured 123 metabolites with CE_TOFMS and 21 metabolites with LC_TOFMS.

We tracked changes and blue font for the location of revisions in the manuscript and also Extended data files. And we corrected all figures more clearly, according to journal format.

REVIEWERS' COMMENTS

Reviewer #1 (Remarks to the Author):

The authors performed experiments showing that threonine extends the lifespan of *C. elegans* through the transcription factors HSF-1 and DAF-16/FOXO that induce the expression of the iron-binding protein ferritin to prevent ferroptosis. The experiments are rigorous and detailed, and the paper is well-organized. My comments from the first round of review were mostly answered, with the exception of fixing Figure 2A that is currently inaccurate. The manuscript is much improved in content, but the English grammar is still very poor and suggestions for improvement are shown below.

Major comment:

Comment 1: Figure 2 panel a is still very unclear (completely wrong) and needs to be corrected. TDH and GCAT are next to the same reaction arrow, which does not make sense. This should be split into separate reactions with the first reaction showing TDH conversion of threonine to 2-amino-3-ketobutyrate, which then either spontaneously decarboxylates into aminoacetone or is metabolized by the GCAT enzyme using CoA as a co-substrate to form glycine and acetyl-CoA. Figure 2a legend needs to be corrected as well.

Lines 260-262 in Figure 2A legend needs to be completely re-written: Threonine is metabolised from glycine or homoserine and catabolized to aminoacetone and methylglyoxal or a branched chain amino acid by the enzyme threonine dehydrogenase (TDH). -> Threonine is metabolized from glycine by threonine aldolase. Threonine is metabolized by threonine dehydrogenase (TDH) to 2-amino-3-ketobutyrate, which then either spontaneously decarboxylates into aminoacetone or is metabolized by the GCAT enzyme using CoA as a co-substrate to form glycine and acetyl-CoA.

Minor comments (wording):

p. 3 line 45: amino acid impairment -> altered amino acid metabolism

p. 4 line 55: involved -> involved in

p. 4 line 57: are enhancing -> enhanced

p. 4 line 60: was no significant alteration -> were no significant alterations

p. 4 line 60: pumping reflected -> pumping rate reflecting

p. 4 line 60: with -> caused by

p. 4 line 61: levels -> activities

p. 4 line 62: enzymes, superoxide dismutase (SOD), catalase activity, and -> enzymes superoxide dismutase (SOD) and catalase, and increased the

p. 4 line 66: or -> and as a primary

p. 4 line 72: R102.4 -> R102.4 (threonine aldolase)

p. 4 line 73: located in -> located at

p. 4 line 77: Day -> day

p. 4 line 78: metabolising -> synthesis

p. 4 line 79: as longevity metabolites -> as a longevity metabolite

p. 5 line 83: by -> of

p. 5 line 88: DR with -> the dietarily restricted

p. 5 line 88: Day -> day

p. 5 line 89: threonine -> threonine levels

p. 5 line 90: healthspan-related factors -> healthspan

p. 5 line 91: Impairment -> In summary, impairment

p. 5 line 91: two different enzyme RNAi clones had adverse effects on lifespan in -> RNAi clones targeting two different enzymes of threonine metabolism altered the lifespan of

p. 5 line 93: model -> this model

p. 5 line 95: transcriptional oxidative stress-response -> oxidative stress-response transcription

p. 5 line 101: given the distinguished role of lifespan -> and how they may regulate lifespan

p. 5 line 103: mutant -> mutants

p. 5 line 103: extend a -> extend the

p. 5 line 107: may because -> may be because

p. 5 line 108: another threonine catabolic enzyme, glycine-C-acetyltransferase (GCAT) -> glycine-C-acetyltransferase (GCAT) that metabolizes the 2-amino-3-ketobutyrate product of the TDH reaction

p. 5 line 111: lacked -> lack

p. 6 line 118: factor -> factors

p. 6 line 119: respective RNAi for worms -> their respective RNAi clone

p. 6 line 120: by applying -> induced by

p. 6 line 139: estimated whether iron contents -> hypothesized that altered iron content

p. 6 line 140: remove the words "rate of"

p. 6 line 141: threonine -> threonine addition

p. 6 line 141: indicated a -> suggested a possible

p. 6 line 142: expended iron retention of -> expanded iron retention by

p. 7 line 147: reduced threonine -> supplemental threonine

p. 7 line 147: increase in ROS -> decrease in ROS

p. 7 line 150: irritate -> stress

p. 7 line 152: With the age-response -> There was an age-related

p. 7 line 152: threonine increasing interventions -> and addition of increasing amounts of threonine

p. 7 line 154: could induce lipid peroxidation, which executed -> has been shown to induce lipid peroxidation, which mediates

p. 7 line 158: raised the hypothesis -> suggested

p. 7 line 160: recovery of -> recovery of wild-type

p. 7 line 160: expressions -> expression

p. 7 line 161: under ftn-1 depletion status -> in ftn-1 mutant nematodes

p. 7 line 162: lessened -> decreased the

p. 7 line 162: in -> in a

p. 7 line 164: provide -> provide

p. 7 line 167: representative -> most representative

p. 7 line 170: aging -> aging,

p. 7 line 178: with -> with a

p. 8 line 180: current -> the current

p. 8 line 181: humans however, -> humans that lack functional threonine aldolase and threonine dehydrogenase enzymes. However,

p. 8 line 184: itself lead -> led

p. 8 line 184: leads through -> led to

p. 8 line 184: leads to led

p. 8 line 184: Move the phrase “through a DAF-16 and HSF-1 dependent mechanism” to the end of the sentence after “C. elegans”.

p. 8 line 185: attenuates -> attenuated

p. 8 line 185: mediates -> mediated

p. 8 line 185: extension and healthspan-contributing factors -> and healthspan extension

p. 9 line 189: most -> the most

p. 9 line 189: limitation of proteins or amino acids -> protein or amino acid restriction

p. 9 line 190: forceful single molecular restriction -> potent singular dietary intervention

p. 9 line 191: is also impacting on -> also impacts

p. 9 line 195: glycin -> glycine

p. 9 line 195: threonine -> the level of threonine

p. 9 line 195: restored on -> restored in

p. 9 line 196: Remove the words "the possibility"

p. 9 line 196: slightly -> that the slightly

p. 9 line 196: glycine -> glycine level

p. 9 line 198: This -> A past

p. 9 line 200: contents -> content

p. 9 line 202: it -> if its increased abundance

p. 9 line 204: many of studies -> many studies

p. 9 line 215: generation -> the generation

p. 9 line 215: indiscriminate reduction of -> its indiscriminate reaction with

p. 9 line 215: peroxides, such as fenton -> peroxides by the Fenton

p. 9 line 218: oxidise -> oxidize

p. 9 line 218: intense -> extreme

p. 9 line 219: upregulates -> upregulate

p. 10 line 220: reduces -> reducing

p. 10 line 220: blocks -> blocking

p. 10 line 224: initiated -> induced

p. 10 line 224: indicated -> indicates

p. 10 line 224: modulation -> levels

p. 10 line 226: case -> the case

p. 10 line 226: diseases -> disease

p. 10 line 229: angle how -> mechanism through which

- p. 10 line 230: further -> stimulate
- p. 11 line 234: Any method -> Methods
- p. 11 line 234: supplemental -> and supplemental
- p. 12 line 250: Please italicize eat-2.
- p. 12 line 252: colour -> color
- p. 12 line 253: speed -> speed of locomotion
- p. 12 line 257: mimic -> mimetic
- p. 12 line 268: K01C8.1 -> K01C8.1 (SRDH-1 (serine/threonine ammonia lyase))
- p. 12 line 272: speed -> speed of locomotion
- p. 12 line 273: size -> length
- p. 13 line 277: Remove the words "transcription factors"
- p. 13 line 304: under the -> with
- p. 13 line 305: by -> to induce
- p. 14 line 310 its -> their
- p. 15 line 329: nematode -> nematodes
- p. 15 line 338: mutant of -> mutant
- p. 15 line 350: grounded frozen pellets -> frozen pellets were ground for lysis
- p. 15 line 352: with condition of -> using a setting of
- p. 15 line 352: rest -> incubated
- p. 15 line 352: 1 min, -> 1 min, and
- p. 16 line 353: supernatant -> the supernatant
- p. 16 line 353: with -> with a
- p. 16 line 354: with -> to
- p. 16 line 357: and -> and a
- p. 16 line 357: of stock -> of the stock
- p. 16 line 358: threonine stock solution sterilized with -> the threonine stock solution was sterilized with a
- p. 16 line 359: in -> in a suspension of
- p. 16 line 360: Why did you use 1M magnesium sulfate? Isn't that concentration too high?

p. 16 line 360: it applied with RNAi clones -> RNAi clones were used. In this case, the bacteria were not heat inactivated.

p. 16 line 367: additionally -> additional

p. 16 line 367: initiated -> was initiated

p. 16 line 368: transferring -> were transferred

p. 16 line 369: transcripts -> transcript

p. 16 line 369: was reduced -> were reduced

p. 16 line 369: on -> on the

p. 16 line 377: of -> off

p. 16 line 378: showed -> those that showed

p. 16 line 379: Kaplan-Meire -> Kaplan-Meier

p. 16 line 383: with -> according to the

p. 16 line 384: pellets -> pellets were placed

p. 16 line 385: samples cool-downed -> samples were cooled-down

p. 17 line 390: grounded -> were ground

p. 17 line 390: A superoxide -> Superoxide

p. 17 line 391: SOD colormetric -> SOD colorimetric

p. 17 line 393: was determined -> were determined

p. 17 line 396: treated each -> were treated for each

p. 17 line 397: repeated for -> were repeated

p. 17 line 398: exposed -> exposed to

p. 17 line 398: survival worms were -> the survival of the worms was

p. 17 line 399: performed -> were performed in

p. 17 line 402: Nematode -> Nematodes

p. 17 line 404: normalized -> normalization

p. 17 line 405: washed -> were washed

p. 17 line 412: Day 10 of -> On day 10,

p. 17 line 412: recorded 30 second movement -> 30 second movements were recorded

p. 17 line 415: measure body size -> body size measurements

p. 17 line 415: washed -> were washed

p. 17 line 416: under -> using a

p. 18 line 420: nematode -> nematodes

p. 18 line 421: counted -> were counted

p. 18 line 429: frozen -> a frozen

p. 18 line 429: sample -> the sample was

p. 18 line 432: protocol -> a protocol

p. 18 line 433: using -> using a

p. 18 line 436: strain was -> strains were

p. 18 line 442: Worms were prepared approximately 500 -> Approximately 500 worms were prepared

p. 19 line 452: were proceed -> proceeded

p. 19 line 454: was -> were

p. 19 line 456: in five -> five

p. 19 line 456: ends up with 35 Mio -> resulted in 35 million

p. 19 line 457: Remove the word "approach"

p. 19 line 467: lon -> Iron

p. 19 line 467: with -> according to the

p. 19 line 467: frozen worm pellets -> worm pellets were frozen

p. 19 line 469: by acidic buffer is -> into the acidic buffer was

p. 19 line 469: proportion -> proportional

p. 19 line 474: Day 10 of -> On day 10,

p. 19 line 474: 100 mM -> 100 μ M

p. 19 line 481: assy kit -> assay

p. 19 line 483: worms -> worms were

p. 19 line 483: with condition -> using a setting

p. 19 line 483: rest -> incubated

p. 19 line 483: 1 min, -> 1 min, and

p. 19 line 484: and -> and the

p. 20 line 485: used for -> was used for the

p. 20 line 486: treated -> were treated

p. 20 line 491: assays -> assay

p. 20 line 491: tests described each experiments -> test described for each experiment

Reviewer #2 (Remarks to the Author):

The authors addressed all my comments, therefore I recommend this manuscript to be accepted for publications.

Reviewer #3 (Remarks to the Author):

Thank you to the authors for responding to my comments, I would like to follow up on 2 specific points that are technical in nature but are nonetheless important:

1. Line 50: it continues to state that LC-MS/MS was used primarily in the metabolomics analysis. This should be corrected accordingly in the light of the response the author has provided.

2. Figure 1A: Assigning metabolites to metabolic pathways can be tricky given that many metabolites are involved in multiple KEGG metabolic pathways. The authors may want to provide details in the method section on how metabolites were assigned to metabolic pathways in the figure. The resolution of Figure 1A in the merged PDF file version provided is poor - it is important that compound names are legible.

Reviewer #1:

Comment 1:

Figure 2 panel a is still very unclear (completely wrong) and needs to be corrected. TDH and GCAT are next to the same reaction arrow, which does not make sense. This should be split into separate reactions with the first reaction showing TDH conversion of threonine to 2-amino-3-ketobutyrate, which then either spontaneously decarboxylates into aminoacetone or is metabolized by the GCAT enzyme using CoA as a co-substrate to form glycine and acetyl-CoA. Figure 2a legend needs to be corrected as well.

First of all, we would like to express our sincere thanks for your constructive comments. We corrected the Figure 2a panel to more clearly. TDH and GCAT were divided into separate reactions and we also corrected Figure 2a legend as well.

Minor Comment (wording):

Thank you for all the indications. We changed every wording in the document to reflect the suggestions made by the reviewer.

Reviewer #2:

Remarks to the Author:

The authors addressed all my comments, therefore I recommend this manuscript to be accepted for publications.

We are grateful for your review and recommendation.

Reviewer #3:

Specific points:

1. Line 50: it continues to state that LC-MS/MS was used primarily in the metabolomics analysis. This should be corrected accordingly in the light of the response the author has provided.

We appreciate and totally agree with the reviewer's comments and corrected the methods of metabolomics analysis as "Using a CE-TOFMS and LC-TOFMS metabolic profiling" in the result section.

2. Figure 1A: Assigning metabolites to metabolic pathways can be tricky given that many metabolites are involved in multiple KEGG metabolic pathways. The authors may want to provide details in the method section on how metabolites were assigned to metabolic pathways in the figure. The resolution of Figure 1A in the merged PDF file version provided is poor - it is important that compound names are legible.

Figure 1a: We assigned metabolites to metabolic pathway using KEGG metabolic pathway. The representative metabolites, which remarkably elevated were depicted in Figure 1b. Because it is important to present alteration of metabolic pathway more specific metabolites in Figure 1a, we displayed as shown and also, specific names of metabolic compounds were provided in the source data of this study. Thank you for your kind advise.

Sincerely yours,

Juewon Kim, Ph.D. and Dongryeol Ryu, Ph. D.